# Generative Perception of Shape and Material from Differential Motion

**Xinran Nicole Han**
Harvard University
xinranhan@g.harvard.edu

**Ko Nishino**
Kyoto University
kon@i.kyoto-u.ac.jp

**Todd Zickler**
Harvard University
zickler@seas.harvard.edu

## Abstract

Perceiving the shape and material of an object from a single image is inherently ambiguous, especially when lighting is unknown and unconstrained. Despite this, humans can often disentangle shape and material, and when they are uncertain, they often move their head slightly or rotate the object to help resolve the ambiguities. Inspired by this behavior, we introduce a novel conditional denoising-diffusion model that generates samples of shape-and-material maps from a short video of an object undergoing differential motions. Our parameter-efficient architecture allows training directly in pixel-space, and it generates many disentangled attributes of an object simultaneously. Trained on a modest number of synthetic object-motion videos with supervision on shape and material, the model exhibits compelling emergent behavior: For static observations, it produces diverse, multimodal predictions of plausible shape-and-material maps that capture the inherent ambiguities; and when objects move, the distributions converge to more accurate explanations. The model also produces high-quality shape-and-material estimates for less ambiguous, real-world objects. By moving beyond single-view to continuous motion observations, and by using generative perception to capture visual ambiguities, our work suggests ways to improve visual reasoning in physically-embodied systems.[1]

## 1 Introduction

A single image of an object is inherently ambiguous. The image can be mimicked by a planar surface with paint, for example, or a curved mirror under suitable lighting. There are many possibilities, and they arise from the intertwined roles of shape and material in image formation. We argue that vision systems should reason about shape and material jointly, and that they should embrace and model their ambiguities, so that additional cues like motion or touch can be triggered for disambiguation. This will help embodied AI systems deal with very practical problems, such as distinguishing a wet floor from a glossy one, or distinguishing a painted visual cliff from an actual cliff.

Humans naturally resolve much of the uncertainty through motion. Imagine moving a shiny textured object under natural lighting: glossy highlights sweep across its surface, immediately revealing its material and geometric structure. This interplay of material, surface, and motion provides far more insight than a single static observation. Just as young children explore their environment by manipulating objects, computational vision systems can benefit from observing differential object motion—even across a few video frames. These observations motivate a perception framework that (*i*) explicitly models shape-material ambiguity rather than collapsing to a single estimate, and (*ii*) leverages object motion as a natural cue for reducing ambiguity when motion is available.

In contrast to these principles, recent learning-based approaches in computer vision typically focus on inferring either shape or material, but not both, and they are often optimized to predict a single

---

[1]Project page: https://xrhan.github.io/diffmotion/

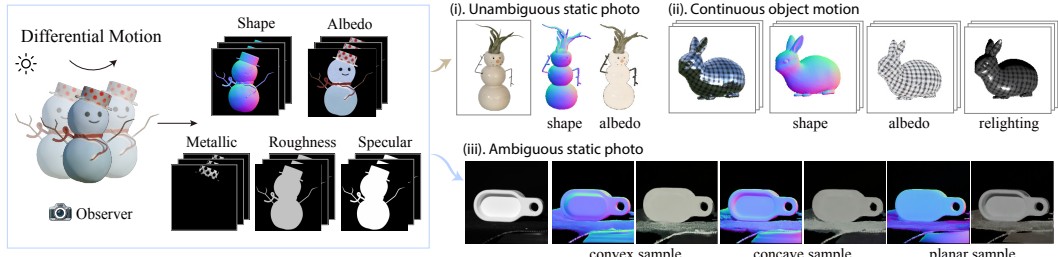

Figure 1: *Left*: We train our generative perception model to *jointly* infer shape and materials using synthetic three-frame videos of objects undergoing differential motions. *Right:* Our model *(i)* generalizes to captured real-world photos; *(ii)* leverages continuous observations to disentangle complex objects, and *(iii)* shows an emergent ability to provide multiple hypotheses for ambiguous static images, such as convex, concave and planar 'postcard' explanations.

"best" output even for tasks that are fundamentally ambiguous. Examples include the estimation of monocular shape (e.g., depth [33, 59] or normals [43, 60, 8]) and intrinsic texture or color [11]. In addition to focusing on a single attribute, such models tend to ignore the ambiguity inherent in the visual world, treating uncertainty as noise to be suppressed rather than something to be modeled.

Instead, we advocate a *generative perception* approach, where a vision system generates diverse samples of shapes and materials that explain the input observation(s). When the input is a single image, the output samples should reflect the full range of plausible interpretations, expressing the fundamental ambiguities of the scene. Enyo and Nishino [18], Kocsis et al. [36], and Han et al. [23] have previously explored this idea for single images and particular attributes, demonstrating generative perception, respectively, of illumination and reflectance; materials; and shape. Here, we argue that generative perception should not be limited to a single view, or to a single object attribute. It should incorporate moving observations when available, and it should jointly disentangle all of shape, texture, and reflectance.

To realize this goal, we introduce a conditional video diffusion model for generative perception that jointly infers shape and material from both static and moving objects. Our denoising diffusion architecture, called **U-ViT3D-Mixer**, operates directly in $256 \times 256$ pixel space and uses spatio-temporal attention and cross-channel mixing to simultaneously reason about surface normals, diffuse texture and reflectance. At coarse scales it employs global attention to capture object-scale constraints, and at fine scales it employs parameter-efficient neighborhood attention to capture local space-time constraints on shape and material.

We train our model from scratch on short synthetic videos of moving and static objects, and we find that it generalizes to captured images and videos while exhibiting several desirable perceptual capabilities: *(i)* it achieves competitive results on existing static-image shape and material benchmarks; *(ii)* it exhibits an emergent ability to generate plausible multimodal samples when the input is ambiguous, such as the classic convex/concave ambiguity or the trivial postcard solution (see Figure 1); and crucially, *(iii)* it makes effective use of differential object motion to resolve perceptual ambiguities, improving prediction accuracy when additional motion frames are available.

Generative perception is a framework for understanding how vision systems can operate under inherent ambiguity. Our work advances this framework by extending from static images to motion videos, and by estimating shape and material together. We hope this opens new directions for cognitive modeling, and for embodied AI systems that aim to emulate human perception.

## 2   Related Work

We study a novel problem: generative disentanglement of object shape and material from short videos. This has not been directly addressed in prior work, but it relates to several research threads.

**Intrinsic image decomposition and inverse rendering**. Intrinsic image decomposition aims to separate an image into texture (albedo), diffuse shading, and sometimes additional reflectance properties such as metallic-ness and roughness, all without inferring shape [7, 36, 11]. Inverse

rendering extends this goal by recovering the full scene properties—including shape, materials, and illumination—that are necessary for physically accurate re-rendering [41, 4, 42, 47, 61]. Our work differs from these by estimating shape and materials without explicit illumination, thereby focusing on all of the scene attributes that are intrinsic to an object.

Recently, Zeng et al. [61] introduced RGB-X, which finetunes latent diffusion models for single-image prediction of surface normals, material properties, and illumination, while also supporting text-conditioned synthesis. The model differs from ours by outputting each attribute independently via a prompt selector, without modeling their interactions. Similarly, DiffusionRenderer [40] finetunes Stable Video Diffusion [9] for physical attributes prediction given video observations. The model conditions on a domain embedding (e.g. "normals", "roughness") to infer one target attribute at a time. In contrast, our model integrates shape and material reasoning within a single backbone, allowing their representations to inform each other.

**Diffusion models for conditional prediction**. Generative modeling with diffusion models has gained increasing recent attention [28, 54, 16, 49]. Many recent works have leveraged priors from pre-trained latent diffusion models for dense prediction tasks such as estimating depth [33, 26], normals [43, 19, 46, 60, 8], or materials [36, 61]. These methods often suppress output diversity by averaging multiple samples [33, 36] or regularizing variance [60], overlooking the inherent ambiguity in the tasks and leading to over-smoothed predictions that fail to capture multimodal outcomes. In contrast, recent *generative perception* models embrace ambiguity by using diffusion models to produce diverse samples of illumination and reflectance [18] or of shape [23] from a single image. Here we extend this idea to consider motion and to infer shape and materials together.

**Shape from differential motion**. Prior mathematical analyses have used differential equations to study the reconstruction of shape from small motions under idealized conditions, such as directional lighting and Lambertian or mirror-like materials [44, 5, 57]. Belhumeur et al. [6] show that for Lambertian shapes with unknown lighting there are continuous stretches and tilts (the generalized bas-relief ambiguity) that cannot be resolved by small motions. When lighting and reflectance (BRDF) are both unknown, Chandraker et al. [12] show that three observations of differential motions around three distinct axes are needed to correctly reconstruct the shape along a characteristic surface curve. Thus, from a mathematical standpoint, perceptual ambiguity persists even when observing differential motion, which highlights the need for generative models that can represent the distribution of possibilities. A learning-based approach like ours also has the benefit of applying to a broader range of lights and materials that would be very difficult to characterize analytically.

# 3 Methodology

We consider a rigid object undergoing small motion with respect to a stationary viewer and lighting environment. The observer is represented by an orthographic camera along the $z$-axis in world coordinates. Given a sequence of temporally continuous RGB image observations $c \in \mathbb{R}^{F \times H \times W \times 3}$ with $F$ frames, we aim to infer time-varying ($F$-frame) *shape-and-material* fields $X = \{n, M = \{\rho_d, r, \gamma, \rho_s\}\}$ from $c$. We use bold symbols to denote variables for multiples frames. Figure 1 depicts our overall approach.

## 3.1 Shape and Material Representations

We represent shape by the viewer-centric surface normal field $n \in \mathbb{R}^{F \times H \times W \times 3}$. The normal vector at the backprojection of each image point $(u, v)$ in a single frame has unit length and is denoted by $n(u, v)$. We represent material attributes $M$ using the point-wise surface albedo (i.e., diffuse color/texture) $\rho_d(u, v)$, roughness $r(u, v)$, metallic-ness $\gamma(u, v)$ and specularity $\rho_s(u, v)$. Each parameter is a time and space varying map with the same dimension $F \times H \times W$ that has either three channels ($\rho_d$) or one channel ($r, \gamma, \rho_s$) and values in the range $[0, 1]$. These parameters are chosen from a subset of those of an analytical reflectance model—the Disney principled BSDF model [10]. Together they determine the bidirectional reflectance distribution function (BRDF)

$$f_r(\omega_i, \omega_o; \{\rho_d, \rho_s, r, \gamma\}) = (1 - \gamma) \frac{\rho_d}{\pi} (f_{\text{diff}}(\omega_i, \omega_o) + f_{\text{retro}}(\omega_i, \omega_o; r))$$
$$+ f_{\text{spec}}(\omega_i, \omega_o; \rho_d, \rho_s, r, \gamma) \, . \tag{1}$$

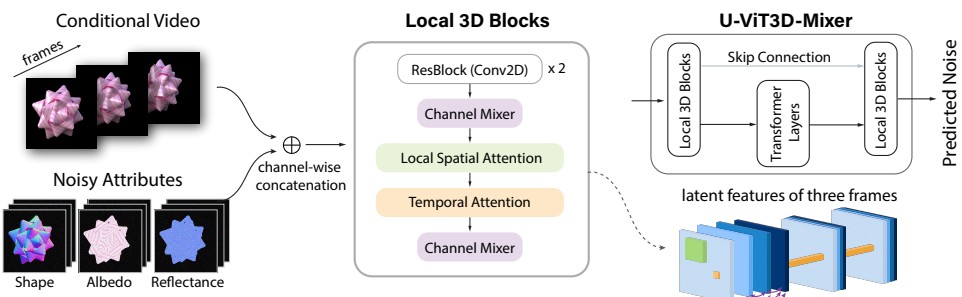

Figure 2: Our parameter-efficient denoising network, U-ViT3D-Mixer, takes in a channel-wise concatenation of conditional video frames and noisy shape-and-material frames. At high spatial resolutions, it uses efficient local 3D blocks (middle) with decoupled spatial, temporal, and channel-wise interactions. At lower spatial resolutions, it uses global transformer layers with full 3D attention.

In an idealized setting without ambient occlusion, this BRDF determines the reflected radiance at each surface point by the physically-based rendering equation [32]

$$L_r(u, v, \omega_o) = \int_\omega f_r(\omega_i, \omega_o) L_i(\omega_i)(\omega_i \cdot n(u, v)) d\omega_i \,, \tag{2}$$

where $\omega_i$ and $\omega_o$ are the incident and outgoing directions of light in the local coordinate frame of the hemisphere centered at normal $n(u, v)$, and $L_i$ is the incident directional illumination. Unlike inverse rendering methods (e.g., [42]), we do not infer lighting and solely focus on recovering normals $n$ and material properties $M$. We provide the full details of the rendering formulation in the appendix.

## 3.2 Generative Perception from Differential Motion

We design a conditional video diffusion model that can exploit the temporal continuity of differential motion and that supports simultaneous stochastic sampling of shape, texture, and reflectance. Our model also processes still images by treating them as static videos.

### 3.2.1 Joint Inference with a Diffusion Model

Our model has the unique feature of predicting shape and materials *jointly* using a single backbone. This design choice is motivated by our preliminary probing experiment (see Appendix A.2), where a model trained to predict albedo can be probed using DPT-style probes as in [17] to produce reasonable estimates of surface normals. This suggests strong cross-attribute interaction and motivates a unified architecture. Our approach is very different from existing models that require fine-tuning multiple pre-trained diffusion models for shape or materials [37], or that use a text prompt to switch between one output attribute at a time [61, 40].

We use a conditional diffusion probabilistic model [28] (DDPM) to generate shape-and-material attributes $x \sim q(X|c)$ by iterative denoising. To train the model we define a 'forward process' that adds Gaussian noise to a clean shape-and-material map $x$:

$$z_t = \alpha_t x + \sigma_t \epsilon, \quad \text{where } \epsilon \sim \mathcal{N}(0, \mathbf{I}) \,. \tag{3}$$

We use a continuous noise schedule where $z_0$ is close to clean data and $z_1$ is close to Gaussian noise, and we use a variance-preserving [28] noise schedule which implies $\alpha_t^2 + \sigma_t^2 = 1$.

We use a neural network with weights $\theta$ to approximate $q(X|c)$ by a tractable distribution $p_\theta(X|c)$. The network learns to estimate the noise in intermediate timesteps $t \in (0, 1)$ for all frames. During training, each conditioning frame $c^f \in c$ is associated with the same noise level $\sigma_t$ and with different, randomly sampled Gaussian noise $\epsilon^f$. We train the network by minimizing the prediction error averaged over all frames:

$$L(\theta) := \frac{1}{|\mathcal{F}|} \sum_{f \in \mathcal{F}} \mathbb{E}_{t \sim \mathcal{U}(0,1), \epsilon^f \sim \mathcal{N}(0, \mathbf{I})} \left[ ||\epsilon^f - \boldsymbol{\epsilon}_\theta(x_t^f, t; c^f)||_2^2 \right] . \tag{4}$$

During sampling, given the conditional video frames $c$, we perform the reverse diffusion process using DDIM [52, 20] beginning with a sample $z_1$ from the standard Gaussian distribution. Given sample $z_t$ at intermediate timestep $t$, we predict the clean sample $\hat{x}$ with the noise estimate $\hat{\epsilon}_\theta$ from the denoiser neural network and project it to a lower noise level $s < t$ using

$$z_s = \alpha_s \hat{x} + \sigma_s \hat{\epsilon}_\theta(z_t, t; c), \quad \text{where } \hat{x} = (z_t - \sigma_t \hat{\epsilon}_\theta)/\alpha_t. \tag{5}$$

As $s \to 0$, we obtain the shape-and-material prediction $\hat{x}^f$ for $f \in \mathcal{F}$. The sampling process of DDIM can be made deterministic and thus the randomness only comes from the initial noise $z_1$.

### 3.2.2 Spatio-temporal and Multi-attribute Interactions

Unlike prior work that fine-tunes large pretrained diffusion models for estimating either shape [33, 43, 19, 46, 60] or materials [61, 7], we train our model to estimate both *from scratch* using a modest set of synthetic videos. This choice is motivated by findings from Han et al. [23], which demonstrate that pretrained models with strong priors can lack the flexibility to provide multimodal output distributions in the presence of ambiguity. Our training strategy, along with an improved noise schedule [35, 30] and our parameter-efficient architecture, allows us to investigate whether multimodal, generative perception can emerge naturally from learning, without being affected by the potential biases in large pretrained models.

As shown in Figure 2, our model takes in a set of input video frames $c^f$ and concatenates it channel-wise with a set of noisy per-frame shape-and-material maps $x_t^f = \{n_t^f, M_t^f\}, f \in \mathcal{F}$. This yields 12 channels per frame as the input to the denoising network. We introduce the following network components that act across spatial, temporal, and attribute dimensions to achieve joint shape-and-material perception from observations of differential motion.

**Spatio-temporal interaction**. Scaling pixel-space diffusion models to operate at higher resolutions (e.g., 256×256) introduces significant computational challenges. For instance, the diffusion UNet [28] and DiT [48] operate with full self-attention in each block, which becomes infeasible at higher resolutions due to a quadratic growth in token count.

To address this, we adopt a hybrid design inspired by U-ViT [29, 30] and Hourglass Diffusion Transformer [14]. At high spatial resolutions, we use ResNet-style convolution blocks [27] followed by separate local spatial attention (via shift-invariant neighborhood attention [25]) and temporal attention layers. At coarser resolutions (e.g., 32×32, 16×16), we apply transformer blocks [48] with global spatio-temporal attention and use 3D Rotary Positional Embeddings [55].

This spatial-temporal interaction at higher resolutions is particularly suited to exploiting the local space-time cues for shape, material and motion that are known to exist mathematically [12]. Our ablations show that it notably improves performance for both shape and material estimation.

**Multi-attribute mixer**. While convolution and local attention modules extract strong spatio-temporal features, they do not promote interaction across channels representing different physical attributes. To address this, we incorporate channel mixer blocks inspired by MLP-Mixer [56] after early convolutional and local attention layers. These blocks enable explicit inter-attribute communication, accelerating convergence and improving predictions.

Since our model performs multi-attribute denoising across shape, albedo and reflectance, we apply attribute-specific loss weighting coefficients $\lambda_{\text{shape}}, \lambda_{\text{albedo}}, \lambda_{\text{reflectance}}$. We find that upweighting shape and albedo ($\lambda_{\text{shape}} = \lambda_{\text{albedo}} = 0.4$) relative to metallic-ness, specular and roughness parameters ($\lambda_{\text{reflectance}} = 0.2$) leads to faster convergence and better overall performance.

In summary, our architecture extends existing pixel-space image diffusion architectures [50, 29, 29, 30, 14] to a video-conditioned, multi-attribute setting, enabling high-resolution generation with rich spatial, temporal, and cross-attribute interactions.

## 4 Experiments

To the best of our knowledge, no existing dataset provides high-quality annotations of physical object attributes under object motion. We therefore construct a synthetic dataset for training using the Mitsuba3 renderer [31] with custom integrators to extract ground-truth shape and material. We source objects from Adobe 3D Assets [1], obtaining 1100 artist-designed models. To enhance diversity,

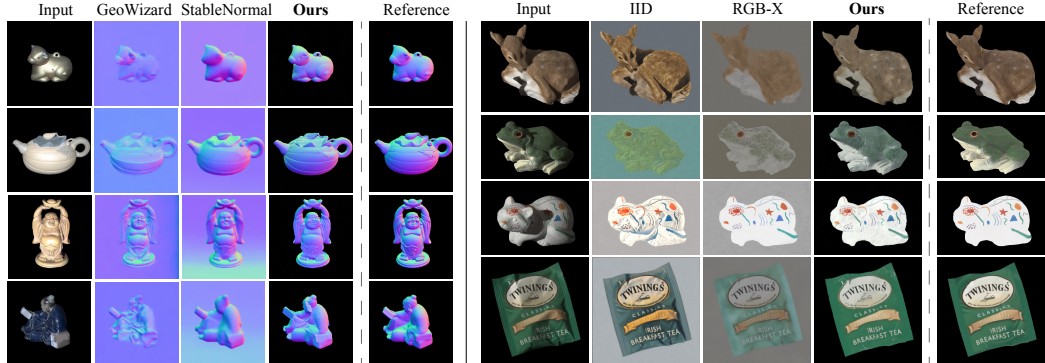

Figure 3: Qualitative comparisons of estimated shape and albedo/texture from a static scene. For fair comparison of scale-invariant albedos, we visualize the scaled albedo from each model closest to the ground truth in the masked object region. Our model achieves comparable shape predictions with existing baselines that specializes at shape prediction, and it achieves better quality in albedo estimation. Our model also produces greater spatial detail.

Table 1: Quantitative evaluation on shape and normal estimation tasks.

| Model | Shape Estimation (DiLiGent) | | Albedo Estimation (MIT-Intrinsic) | | |
| | Mean AE ($\downarrow$) | Median AE ($\downarrow$) | SSIM ($\uparrow$) | RMSE ($\downarrow$) | PSNR ($\uparrow$) |
|---|---|---|---|---|---|
| Marigold-E2E-FT [46] | **17.74** | 13.86 | - | - | - |
| StableNormal [60] | 18.21 | 14.59 | - | - | - |
| DSINE [3] | 20.29 | 16.17 | - | - | - |
| GeoWizard [19] | 24.62 | 21.55 | - | - | - |
| IID [36] | - | - | 0.61 | 0.13 | 18.19 |
| ColorfulShading [11] | - | - | 0.72 | 0.10 | 20.56 |
| Marigold–Albedo [34] | - | - | 0.72 | 0.11 | 19.86 |
| RGB-X [61] | 34.99 | 32.10 | 0.71 | 0.11 | 20.82 |
| Ours | 18.03 | **12.31** | **0.80** | **0.08** | **23.14** |

each shape is rendered under two material settings: (*i*) its original artist-designed materials, and (*ii*) a randomly sampled texture from a set of 400 open-source assets [13, 15, 58] combined with spatially-uniform reflectance from randomly sampled metallic, roughness, and specular coefficients.

All scenes are lit with one of 200 hundred randomly-sampled HDR environment maps collected from [2, 21], with horizontal and vertical environment flips for augmentation. To simulate object motion, we place the object centroid in the image center, randomly sample a 3D rotation for initial pose, and apply a motion rotation about the vertical image axis with a random angle sampled from $\{[-8°, -2°], 0°, [2°, 8°]\}$ with probability $[0.4, 0.2, 0.4]$. This means that $20\%$ of training samples are static scenes containing no motion. We generate 45 short video clips (5 frames each) for each object, and we split them into pairs of consecutive 3-frame clips ($F = 3$) for training. This results in a dataset of approximately 100K video-attribute pairs.

We train our model using the AdamW optimizer [45]. Training requires roughly five days using four H100 GPUs. We follow the sigmoid loss reweighting strategy [35, 30] and adopt a shifted-cosine continuous noise schedule [30, 53]. Complete hyperparameter settings for our architecture are provided in the appendix. During inference, we use DDIM sampling [52] with 50 steps, which takes about 2.7 seconds per input video on a single A100 GPU.

## 4.1 Static Object Experiments

Our model processes single images of static objects by using $F$ image copies as input, so we can assess its shape and material performance using existing single-image datasets. We evaluate our model in two ways: (*i*) its similarity to ground truth for everyday objects that are less ambiguous, and (*ii*) its ability to generate plausible multimodal solutions when the input is more ambiguous.

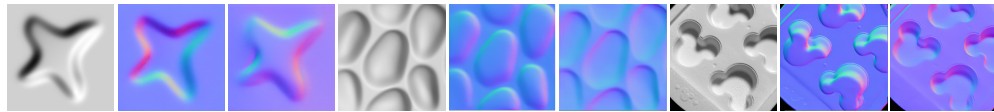

Figure 4: Our model exhibits multimodal shape perception on ambiguous visual stimuli presented in Han et al. [23] despite only being trained on everyday objects and without specific data augmentation.

Note that all comparison models in this section except RGB-X [61] are specialized for either shape or material estimation. Also, we cannot compare to the previous generative perception approaches of [18, 23] because they require either known shape or textureless diffuse objects.

We evaluate shape accuracy using the DiliGENT dataset [51]. This dataset contains photos taken under point light sources, originally designed for the photometric stereo task. Notably, this lighting is quite different from the environment lighting in our training set. For each of the 10 objects, we randomly sample 5 images and compute the mean and median angular surface normal errors within the masked object region. We compare with state-of-the-art shape prediction methods: StableNormal [60], Marigold-E2E-FT [46], GeoWizard [19], and DSINE [3]. All except DSINE are fine-tuned from pre-trained image diffusion models. For stochastic methods (ours, StableNormal, GeoWizard), we report the average of the three lowest error values from among 10 predictions.

We assess albedo/texture estimation using the MIT Intrinsic Image Dataset [22] which provides object photos under laboratory lighting and ground-truth albedo maps. We compare to a deterministic method, ColorfulShading [11], and diffusion-based methods Intrinsic Image Diffusion (IID) [36], Marigold–Albedo [34] and RGB-X [61]. Following prior work [36, 11], we use scale-invariant SSIM, RMSE, and PSNR for evaluation. For the IID model, we follow their recommendation by using the mean of 10 samples. For RGB-X, Marigold–Albedo and our model, we report the average of the three lowest error values from 10 predictions.

Table 1 shows quantitative results, and Figure 3 shows qualitative comparisons with other diffusion-based methods. For shape estimation, our model performs competitively with existing methods that specializes in shape prediction. Note that our model estimates the material in addition to the shape and is trained solely on synthetic data. For albedo estimation, our model outperforms existing models by a large margin. We attribute this improvement to (*i*) the use of pixel-space diffusion, which helps preserve fine details, and (*ii*) our synthetic training dataset with diverse and challenging objects using randomized texture and reflectance.

Overall, our model matches or exceeds all of the specialized shape or material models on existing benchmarks. The only other model that estimates both is RGB-X, but we find that it often struggles with complex shapes and provides less competitive performance across diverse objects.

**Perceptual ambiguity**. Han et al. [23] introduced a set of test images designed to probe the ambiguity-awareness of computational models, particularly for the convex/concave ambiguity. When evaluated on these images and as shown in Figures 1(*iii*) and 4, our model produces multimodal predictions, similar to human perception. Interestingly, due to training on textured rather than homogeneous objects, our model also occasionally exhibits a 'postcard' interpretation, perceiving a flat surface with painted texture. We find the joint estimation of shape and material further enhances interpretability through disentanglement and can offer insight into the model's perceptual hypotheses.

### 4.2 Moving Object Experiments

We compare three ablated variants of our model: (*i*) **U-ViT3D (base)** is a baseline with convolutional layers at high resolutions and transformer blocks in the bottleneck; (*ii*) **+ Local Spatio-Temporal Attention** replaces one ResNet block per resolution level with a local spatial attention and a temporal attention module; and (*iii*) **+ Multi-Attribute Mixer** further adds lightweight channel mixers to each local 3D block.

We conduct ablation studies on a held-out synthetic evaluation set containing objects undergoing differential motion. These are generated using the same pipeline as our training data but with distinct objects to ensure generalization. We evaluate performance under both moving and static observations. For each video, we draw 10 samples and report the mean of the top-3 based on a joint ranking of

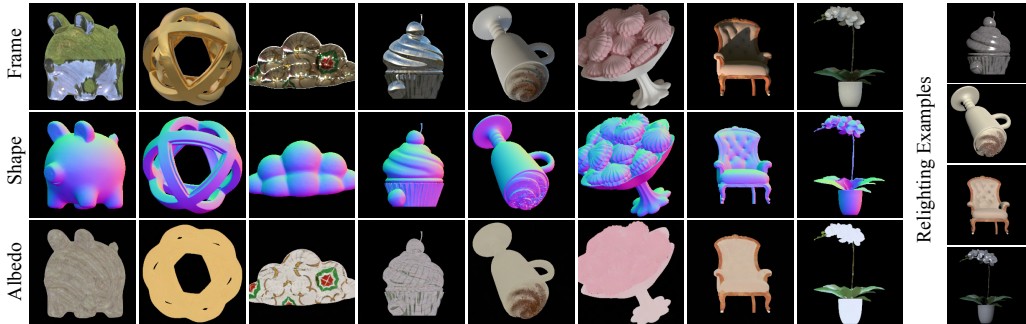

Figure 5: Shape and material estimates from our full model. For each three-frame test video, we show one representative frame. The first four columns use randomly sampled materials, and the last four use original materials. Note that some specular objects are quite challenging; we demonstrate in the supplementary videos how motion aids disambiguation. On the right we show relighting examples using the estimated shape, albedo, and reflectance parameters under directional lighting.

Table 2: Ablation study on shape and albedo estimation using our differential-motion evaluation set. Reported metrics are computed over mean of the top 3 of 10 stochastic samples per video. Model variants are compared under both moving and static observation conditions.

| Model Variant | # Params | With Motion | | Without Motion | |
|---|---|---|---|---|---|
| | | Mean AE $\downarrow$ | SSIM $\uparrow$ | Mean AE $\downarrow$ | SSIM $\uparrow$ |
| U-ViT3D (base) | 95.7M | 18.757 | 0.778 | 22.471 | 0.760 |
| + Local Spatio-Temporal Attention | 97.4M | 14.555 | 0.825 | 18.957 | 0.811 |
| + Multi-Attribute Mixer | 100.3M | **11.262** | **0.832** | **16.886** | **0.823** |

shape (Mean Angular Error (MAE)) and albedo (SSIM). This combined metric helps capture overall performance instead of focusing solely on shape or materials.

Table 2 shows that each module leads to notable performance gains with minimal added parameters. Specifically, we find that using spatial neighborhood attention is crucial for better shape estimation. The base model with only convolutions tends to produce over-smoothed results on complex inputs and often fails to reason about the shape of highly specular objects. Our model also shows better performance when it observes objects under motion instead of being static, highlighting its ability to leverage motion cues. Figure 5 visualizes predictions from the full model on motion inputs.

**Perceptual ambiguity**. To assess our model's ability to adapt its hypotheses in the presence of motion, we apply it to the perceptual demonstrations from Hartung and Kersten [24][2]. These are videos of moving objects rendered with either (a) a uniform shiny material that reflects the environment, or (b) a spatially-varying matte material representing painted texture. The two scenarios are indistinguishable in the absence of motion, because they can produce individual frames that are identical.

We probe our model's interpretations by visualizing its distribution of generated albedo samples, because the shiny interpretation implies a spatially-uniform albedo whereas the matte interpretation does not. The left of Figure 6 shows an embedding of 100 albedo predictions when seeing a shiny teapot that is either static or moving. The albedo samples are diverse in the static case, containing both shiny and matte interpretations, and they converge to the shiny explanation when there is motion.

The right of Figure 6 shows albedo samples for a single video frame that is common across two distinct videos of the same shape rendered as shiny or matte. The frame is ambiguous in isolation, so the interpretation is heavily influenced by temporal motion. We find that our model clearly distinguishes between the shiny and matte explanations, with samples forming two distinct clusters.

These experiments demonstrate that our model can effectively leverage differential motion to refine its material prediction, similar to human observers [24]. Note that our dataset does not contain any objects with painted textures that are purposefully designed to look like reflected environments. Nonetheless, it learns to include that possibility among its predictions for static objects, and it learns to either confirm or deny that possibility in the presence of motion.

---

[2]We encourage viewing of videos at `https://kerstenlab.psych.umn.edu/demos/motion-surface/`

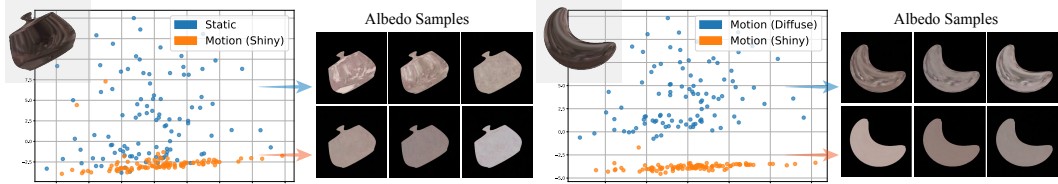

Figure 6: Motion disambiguation results for the Shiny or Matte Ambiguity [24]. Each plot shows a PCA embedding of 100 albedo samples corresponding to the frame (inset) that is common between two distinct input videos. *Left:* When a shiny teapot moves, the albedo samples converge to a *subset* of the ones produced for the static teapot. *Right:* The albedo samples for a matte-rendered motion video are clearly *separated* from those of a shiny-rendered video. Note that spatially-uniform albedos form tighter clusters in PCA space, while highly textured ones exhibit greater variation.

### 4.3 Real-world Examples

Figure 7 and 8 shows additional examples with online images and real-world videos. For each input, we segment the objects and mask the background before passing it to the model.

Figure 7 illustrates how our model jointly interprets shape and albedo from static inputs. In the top row, the carpet uses distorted textures and shading to mimic a 'visual cliff'. In the bottom row, our model generalizes to an artist's painting, producing either a flat 'postcard' interpretation or a shaded 3D one. When the shape is perceived as three-dimensional, the predicted albedo appears more uniform.

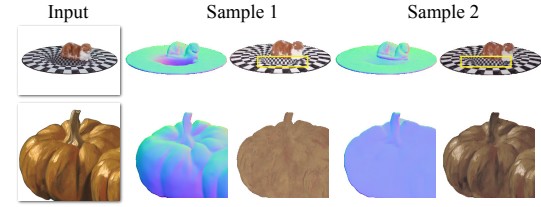

Figure 7: Multiple shape and material hypotheses on an ambiguous online image. A cat on an illusion carpet painted to resemble a hole. When interpreted as a hole (Sample 1), the predicted albedo is brighter in the center region (yellow box) than when interpreted as a plane (Sample 2).

Figure 8 compares our model to RGB-X [61] and DiffusionRenderer [40] on real-world videos of various objects under natural outdoor or indoor lighting. We test them in two settings (a) moving objects, same as our training paradigm and (b) moving camera, which is out-of-distribution for our model. For the latter we use the catured images in the Stanford-ORB dataset [38], by feeding our model sets of three images from nearby camera views. In both scenarios, we show that our model produces better visual quality for both shape and albedo, especially for shiny and metallic objects.

To test the overall generalization of our model to unseen motions, we include additional quantitative results in the appendix A.6 and A.7.

**Summary**. Experimentally we find that our model generalizes quite well to out-of-distribution inputs, including to ambiguous perceptual stimuli (Fig. 4) and to captured videos that contain multiple objects or novel types of motions (Fig. 8). We attribute this to: (*i*) our use of local, shift-invariant attention at high resolutions, which helps avoid overfitting to global object features; and (*ii*) our training set, which includes 3D assets composed of multiple objects and self-occluding object-parts.

**Limitations**. Our model currently focuses on restricted classes of motions and materials. While it can handle multiple objects undergoing the same rigid motion, it does not apply to scenes that contain deforming objects, or that contain multiple distinct motions. Nor does it apply to objects that exhibit translucency or transparency. Another limitation of our model is its reliance on objects being pre-segmented from the background. Extending the model to perform perceptual grouping in addition to inferring object attributes is an important future direction.

## 5 Conclusion

Inspired by the interdependence of shape and material perception, and by the human ability to resolve ambiguities through motion, we introduce a generative perception approach that produces combined samples of shape, texture, and reflectance which explain a short video of differential object motion. Our model recovers complete descriptions of an object's shape and materials while capturing and modeling inherent ambiguities. We show that it successfully maintains a distribution

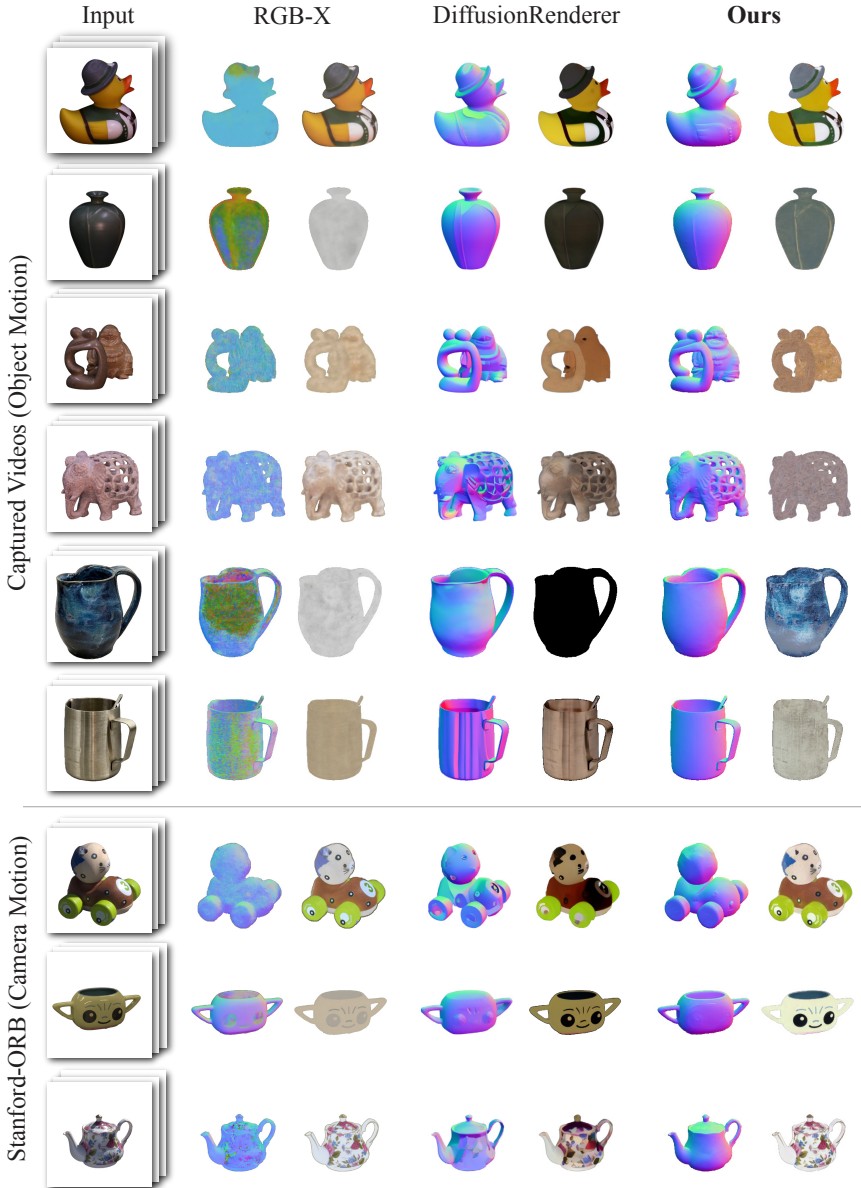

Figure 8: Outputs from captured videos of real world objects. Our model effectively uses differential motion to produce higher-quality estimations of shape and albedo compared to RGB-X and DiffusionRenderer.

of plausible interpretations under ambiguous static observations, and that it naturally converges to more concentrated and accurate hypotheses in the presence of motion. By introducing an effective conditional video diffusion model that operates directly in pixel space, we also provide a potential new backbone for space-time generative perception. We hope this inspires future research in ambiguity-aware perception and dynamic vision for embodied systems.

## Acknowledgments

We thank Jianbo Shi for reviewing the manuscript. We also thank Kohei Yamashita for guidance about the data rendering pipeline and Boyuan Chen for discussion about video diffusion modeling. This work was supported in part by the NSF cooperative agreement PHY-2019786 (an NSF AI Institute, http://iaifi.org) and by JSPS 21H04893 and JST JPMJAP2305.

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

# A Technical Appendices and Supplementary Material

## A.1 Additional Related Work on 3D Reconstruction

Many recent learning-based multi-view scene reconstruction methods infer point clouds from images [71, 69, 70, 68]. These models are designed for deterministic reconstruction instead of modeling ambiguity, and they do not estimate material properties, so often have the surrounding illumination baked into their output point colors. Although some view-synthesis techniques based on NeRFs [67] and Gaussian splatting [63, 62] can disentangle reflectance and lighting, they require many viewpoints and per-scene optimization. In contrast, our method is feed-forward, and it leverages differential motion in short videos to successfully resolve texture–lighting ambiguity without the need for many wide-baseline images and explicit camera pose estimation.

Some other recent studies [43, 64, 71] generate 3D assets from text or a conditioning image, including the generation of object parts that are not visible in the input. These models are often trained on artist-designed, cartoon-style 3D assets instead of photorealistic ones, and their output is typically a mesh plus a texture that has highlights and other illumination effects baked in.

## A.2 Probing experiment on normal estimation from albedo prediction

We conduct a probing experiment with a low-resolution (64×64) diffusion UNet that is trained to infer the object albedo given a conditional image. We are interested in whether the model learns useful representations for predicting object surface normals. To do this, inspired by prior work [17], we first pre-train the albedo prediction UNet and then insert probes similar to the DPT decoder [66] using multiscale convolutions at intermediate layers. We train the lightweight probes to estimate shape from latent features using a small dataset of 10K ground-truth pairs for five epochs. At test time, we directly read out from the inserted probes for surface normal estimates.

As shown in Figure 9, the model trained for albedo prediction can indeed produce plausible shape estimates through probing. As a result, we hypothesize that building a unified framework for both shape and material estimation can leverage a shared representation and enable a more computationally efficient architecture.

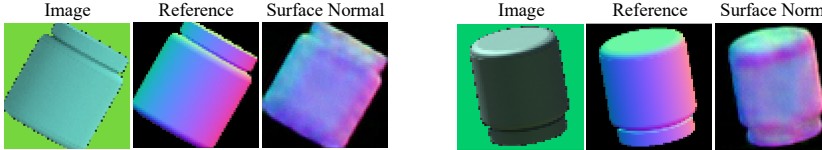

Figure 9: A model trained to predict surface albedo learns representation useful for estimating surface shapes. We show two examples with the ground truth shape (middle) and the readout shape from simple convolutional probes (right).

## A.3 Ablation Studies

In Figure 10a, we compare the shape estimates from our base model (U-ViT3D) with those from our full model (U-ViT3D-Mixer). The base model uses only convolutions at high resolutions, so it tends to produce over-simplified geometry estimates. On the other hand, our full U-ViT3D-Mixer model captures finer details robustly, even for shiny objects.

In Figure 10b, we show the benefits of leveraging motion cues for material estimation. Texture estimation is highly ambiguous when the object is static: the appearance could come from painted texture, a reflected environment, or a mixture of the two. Figure 10b shows samples of our model's textures from ambiguous static images, and it shows how object motion allows the model to separate texture more accurately.

## A.4 Extension using Temporal Consistency Guidance

Our model generates short clips ($F = 3$ frames), but it can be used to create longer sequences by simultaneously generating multiple clips and using inference-time guidance (e.g., [39, 23]) to enforce

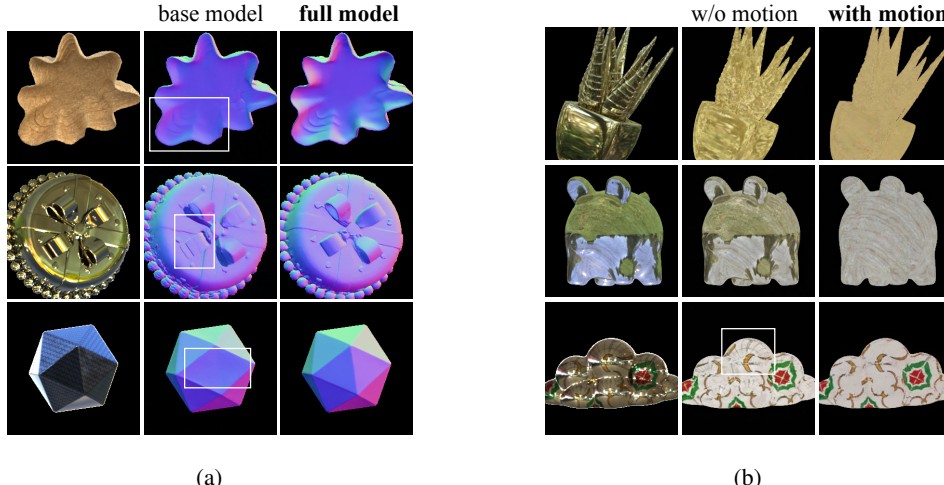

(a)                                                    (b)

Figure 10: *Left*: Ablation of model components. The base U-ViT3D model, i.e., local attention and channel mixing layers ablated from our full model, produces over-smoothed results and struggles with specular surfaces due to complex inter-reflections. *Right*: Ablation of motion from input images. When the shiny objects are rendered as being static, the model can sometimes provide inaccurate estimates due to the inherent ambiguity between texture and illumination. Once the model observes the object undergoing motion, this ambiguity is resolved and the model produces more accurate albedo estimates.

temporal consistency between them. During inference, we apply a reconstruction loss (e.g., MSE loss) at one overlapping frame across adjacent temporal windows, encouraging the denoised predictions $\hat{x}$ to remain consistent at that frame. We find that this can provide temporally consistent shapes and materials over long time horizons, without the need for modifications or additional training. On the project page, we compare our long-horizon results to StableNormal [60], which shows substantially more temporal flickering and inconsistency.

## A.5   Additional evaluation results for IID [36]

In the experiment section Table 1, we use the mean prediction for the IID baseline following the authors' recommended setting because it achieves the best performance in their original paper [36]. The evaluation results with the alternative top-3 out of 10 evaluation protocol are below in Table 3. They are consistently worse than the mean prediction, consistent with the original paper's claims.

Table 3: IID [36] performance under different evaluation protocols.

| IID - Albedo Estimation | SSIM ↑ | RMSE ↓ | PSNR ↑ | LPIPS ↓ |
|---|---|---|---|---|
| Mean prediction out of 10 samples | 0.61 | 0.13 | 18.19 | 0.16 |
| Best 3 out of 10 samples | 0.44 | 0.15 | 17.01 | 0.20 |

## A.6   Quantitative Results using the Stanford-ORB dataset

In the main paper we show qualitative results using images from the Stanford-ORB benchmark [38], where we feed our model sets of three images from nearby camera views. The effective "motion" in this experiment is very different from our training data because (*i*) the camera moves instead of the object; and (*ii*) the camera positions are separated by wide, irregular baselines. Despite this, we find that our model achieves high accuracy, especially in comparison to the two strongest comparison models—Marigold-e2e-ft [46] for shape and ColorfulShading [11] for albedo—using the same input images. Tables 4 and 5 show results for six of the benchmark's scenes, where we see competitive performance for shape and superior performance for albedo. This is an example of our model generalizing beyond the horizontal object motion used for training.

Table 4: Stanford ORB real-world objects: shape inference results

| Model | Shape (Median AE) ↓ | | | | | |
|---|---|---|---|---|---|---|
| | Blocks | Cactus | Car | Cup | Grogu | Teapot |
| Marigold-e2e-ft [46] | 10.19 | 14.59 | 13.69 | 12.14 | 13.27 | 13.25 |
| Ours | 11.22 | 19.50 | 13.86 | 7.85 | 10.84 | 15.02 |

Table 5: Stanford ORB real-world objects: albedo inference results

| Model | Albedo (SSIM ↑ / PSNR ↑) | | | | | |
|---|---|---|---|---|---|---|
| | Blocks | Cactus | Car | Cup | Grogu | Teapot |
| ColorfulShading [11] | 0.76 / 24.69 | 0.66 / 25.58 | 0.64 / 23.01 | 0.84 / 24.83 | 0.61 / 19.87 | 0.69 / 23.33 |
| Ours | 0.74 / 25.28 | 0.78 / 26.23 | 0.70 / 23.68 | 0.88 / 24.97 | 0.86 / 24.05 | 0.79 / 24.12 |

## A.7 Generalization to random rotation and translation

Some examples in the main paper already suggest that our model can generalize to non-horizontal object motions. For instance, in the left video of Figure 6, the teapot rotates vertically, orthogonal to the motion seen during training. To examine this, we perform a simulation experiment using 174 unseen objects with randomized textures, each rendered under three different motion settings:

1. a vertical-axis rotation, as used for training;
2. a rotation about a randomly-chosen axis; and
3. a translation of up to 8% of the object size in a random direction.

Note that (2) and (3) are very different from training, and that (3) is equivalent to having a static image with translational misalignments between frames.

The results in Table 6 show there is limited degradation from in-distribution motions to out-of-distribution ones. We attribute this to the fact that local shape and material constraints derive from the relative angles between rotation axis, normals and curvatures (see [12]), and that since we use a wide variety of shapes during training, the model sees many of these local relationships, each affected by a local translation that depends on the point's 3D position with respect to the rotation axis.

Table 6: Evaluation across different types of object motions.

| Scenario | Shape | | Albedo | |
|---|---|---|---|---|
| | Mean AE ↓ | Median AE ↓ | SSIM ↑ | RMSE ↓ |
| In-dist. vertical-axis rotation | 12.267 | 10.415 | 0.868 | 0.051 |
| Out-of-dist. random rotation | 15.062 | 13.158 | 0.863 | 0.052 |
| Out-of-dist. random translation | 16.652 | 14.681 | 0.858 | 0.054 |

## A.8 Reflectance Model

We describe the full reflectance model based on the Disney principled BSDF model [10]. Recall the reflectance equation

$$f_r(\omega_i, \omega_o; M) = (1 - \gamma) \frac{\rho_d}{\pi} \left[ f_{\text{diff}}(\omega_i, \omega_o) + f_{\text{retro}}(\omega_i, \omega_o; r) \right] + f_{\text{spec}}(\omega_i, \omega_o; \rho_d, \rho_s, r, \gamma), \quad (6)$$

where $M = \{\rho_d, \rho_s, r, \gamma\}$ consists of the material parameters: diffuse albedo $\rho_d$, specular term, $\rho_s$ roughness $r$, metallic-ness coefficient $\gamma$ and $n$ denotes the spatially varying surface normal at a point.

The diffuse term depends on the angle of incident light $\theta_i$ and outgoing direction $\theta_o$:

$$f_{\text{diff}} = \left(1 - F_i/2\right)\left(1 - F_o/2\right), \quad \text{where } F_i = \left(1 - \cos\theta_i\right)^5, \quad F_o = \left(1 - \cos\theta_o\right)^5. \quad (7)$$

To capture retro-reflective highlights, we add

$$f_{\text{retro}} = R_R\left(F_i + F_o + F_i F_o \left(R_R - 1\right)\right), \quad \text{where } R_R = 2\gamma \cos^2\theta_d, \quad \cos\theta_d = h \cdot \omega_i, \quad (8)$$

with $h$ denoting the half-vector between the incident and outgoing directions.

The specular reflection is based on the microfacet model with GGX distribution:

$$f_{\text{spec}} = \frac{F\,D\,G}{4\,(\omega_i\cdot n)\,(\omega_o\cdot n)} = \frac{F\,D\,G}{4\,\cos\theta_i\cos\theta_o}. \tag{9}$$

Here, $D$ is a microfacet distribution function defined by the roughness parameter $r$,

$$D(h) = \frac{r^4}{\pi\big((n\cdot h)^2(r^4-1)+1\big)^2}, \tag{10}$$

and G denotes the masking-shadowing function

$$G = G_1(\omega_i)\,G_1(\omega_o), \text{ where } G_1(\omega) = \frac{2}{1+\sqrt{1+r^4\,(1-n\cdot\omega)^2/(n\cdot\omega)^2}}. \tag{11}$$

The Fresnel term $F$ blends dielectric and metallic responses:

$$F = (1-\gamma)\,F_{\text{dielectric}} + \gamma\,F_{\text{Schlick}}, \tag{12}$$

with

$$F_{\text{dielectric}} = \frac{1}{2}\Big((\frac{\cos\theta_i-\eta\cos\theta_t}{\cos\theta_i+\eta\cos\theta_t})^2 + (\frac{\cos\theta_t-\eta\cos\theta_i}{\cos\theta_t+\eta\cos\theta_i})^2\Big), \text{ where } \eta = \frac{2}{1-\sqrt{0.08\,\rho_s}}-1, \tag{13}$$

where $\theta_t$ is the angle between the normal and the transmitted ray computed using Snell's Law and

$$F_{\text{Schlick}} = \rho_d + (1-\rho_d)\,(1-\cos\theta_d)^5. \tag{14}$$

## A.9    Network Architecture Details

We use the following hyperparameters for the U-ViT3D-Mixer model.

```
channels       = [96, 192, 384, 768],
block_dropout = [0, 0, 0.1, 0.1],
block_type     = ['Local3D'(1), 'Local3D'(1), 'Transformer'(3), 'Transformer'(8)],
noise_embedding_channels = 768,
attention_num_heads = 6,
patch_size = 2,
local_attention_window_size = 7,
channel_mixer_expansion_factor = 3,
loss_type = v-prediction (MSE)
```

We use the following training setup.

```
batch_size = 64,
optimizer = 'AdamW',
adam_betas = (0.9, 0.99),
adam_weight_decay = 0.01
learning_rate = 1e-4,
mixed_precision = 'bfloat16',
max_train_steps = 400k
```

## A.10    Inference Hardware Requirement

We compare our model to other stochastic, diffusion-based baseline models in terms of inference time GPU requirement and runtime. We use each model's FP32 version and conduct the analysis on a single A100 GPU. As shown in Table 7, our model is compact, fast and memory efficient, due to its architectural design.

Table 7: Model comparison of inference requirements.

| Model | Output | Model Size (FP32) | Inference Time | Inference GPU Memory |
|---|---|---|---|---|
| **Ours** | Shape and Materials, three frames | 0.4 GB | 2.7 s | 2.8 GB |
| StableNormal | Shape only, single-frame | 3.8 GB | 1.1 s | 19.5 GB |
| IID | Materials only, single-frame | 6.2 GB | 6.0 s | 9.5 GB |
| RGB-X | Shape *or* Materials, single-frame | 3.8 GB | 10.5 s | 6.0 GB |

# Appendix References

[62] Yingwenqi Jiang, Jiadong Tu, Yuan Liu, Xifeng Gao, Xiaoxiao Long, Wenping Wang, and Yuexin Ma. Gaussianshader: 3D gaussian splatting with shading functions for reflective surfaces. *arXiv preprint arXiv:2311.17977*, 2023.

[63] Zhihao Liang, Qi Zhang, Ying Feng, Ying Shan, and Kui Jia. Gs-ir: 3D gaussian splatting for inverse rendering. *arXiv preprint arXiv:2311.16473*, 2023.

[64] Minghua Liu, Chao Xu, Haian Jin, Linghao Chen, Mukund Varma T, Zexiang Xu, and Hao Su. One-2-3-45: Any single image to 3D mesh in 45 seconds without per-shape optimization. *Advances in Neural Information Processing Systems*, 36, 2024.

[43] Xiaoxiao Long, Yuan-Chen Guo, Cheng Lin, Yuan Liu, Zhiyang Dou, Lingjie Liu, Yuexin Ma, Song-Hai Zhang, Marc Habermann, Christian Theobalt, et al. Wonder3D: Single image to 3D using cross-domain diffusion. In *Proceedings of the IEEE/CVF Conference on Computer Vision and Pattern Recognition*, pages 9970–9980, 2024.

[66] René Ranftl, Alexey Bochkovskiy, and Vladlen Koltun. Vision transformers for dense prediction. In *Proceedings of the IEEE/CVF international conference on computer vision*, pages 12179–12188, 2021.

[67] Dor Verbin, Peter Hedman, Ben Mildenhall, Todd Zickler, Jonathan T. Barron, and Pratul P. Srinivasan. Ref-NeRF: Structured view-dependent appearance for neural radiance fields. In *Proceedings of the IEEE/CVF Conference on Computer Vision and Pattern Recognition*.

[68] Jianyuan Wang, Minghao Chen, Nikita Karaev, Andrea Vedaldi, Christian Rupprecht, and David Novotny. VGGT: Visual geometry grounded transformer. In *Proceedings of the IEEE/CVF Conference on Computer Vision and Pattern Recognition*, 2025.

[69] Qianqian Wang, Yifei Zhang, Aleksander Holynski, Alexei A Efros, and Angjoo Kanazawa. Continuous 3D perception model with persistent state. *arXiv preprint arXiv:2501.12387*, 2025.

[70] Shuzhe Wang, Vincent Leroy, Yohann Cabon, Boris Chidlovskii, and Jerome Revaud. DUSt3R: Geometric 3D vision made easy. In *CVPR*, 2024.

[71] Jianfeng Xiang, Zelong Lv, Sicheng Xu, Yu Deng, Ruicheng Wang, Bowen Zhang, Dong Chen, Xin Tong, and Jiaolong Yang. Structured 3D latents for scalable and versatile 3D generation. *arXiv preprint arXiv:2412.01506*, 2024.

