# OpenReview forum: "Generative Perception of Shape and Material from Differential Motion"
_NeurIPS.cc/2025/Conference — NeurIPS 2025 poster_

### Official Review · Reviewer_yYrM · 2025-06-16

**Clarity:** 3
**Significance:** 3
**Originality:** 3
**Rating:** 5
**Confidence:** 4

**Summary:**

This paper introduces a generative model for the estimation of shape and material parameters from a single image or from a short video sequence. The method relies on a 3D diffusion transformer model, which can diffuse the material properties in a generative fashion (multiple possible outputs from the same input). The results of the paper show that the method is less uncertain about its predictions once a short video sequence is provided, in contrast to using a single image as input. The method is evaluated on synthetic and real-world data, and compared against previous methods and baselines, showing the increased effectiveness of the method with respect to previous work.

**Questions:**

- What is the impact of the input resolution in the output? Can this model be used on higher resolution inputs?
- What motivated the choice of the Disney BRDF? Would other BRDFs work worse?
- Could the authors provide more insights on why the loss weighting coefficients were chosen in this way? My experience is that these inverse rendering algorithms are usually really sensitive to these weightings, and I am curious about these weights, as they are somewhat different to other weights presented in the literature.
- Could the model be more "active" in its exploration? Eg, choose how to rotate the camera, thee objects or the light source. This would would better emulate how humans explore material appearance.

**Ethical Concerns:**

["NO or VERY MINOR ethics concerns only"]

**Final Justification:**

The authors have addressed all of my concerns during the rebuttal. This paper is interesting to the computer vision community, and, while it is not free from important limitations, I think it merits acceptance.

It would still have been benefitial to incorporate real objects, more realistic material models, and more active exploration into the model capabilities and training data, but these are shared limitations of most related work. Nevertheless, the paper is valuable and the incorporation of perceptual metrics really adds value to it, making its analyses much more solid.

**Limitations:**

Yes

**Paper Formatting Concerns:**

No concerns

**Quality:**

3

**Strengths And Weaknesses:**

Strenghts:
- This paper tackles a very important problem in computer graphics and vision, and does it in an ambitious way. I believe these findings can spark further work in the literature.
- The paper is generally well written (with some exceptions), it is very well motivated and it is easy to follow.
- The proposed method is sound, and makes sense in the context of the problem. Learning 3D object representations with transformers under few-shot scenarios is a really difficult task and the proposed diffusion model make sense and is smart. Data generation and augmentation is also done in a smart way.
- The paper provides code, which should enhance reproducilibility.
- Supplementary material provides additional videos and results, which enhance the understanding of the paper, as still images cannot fully convey the domain tbis method is operating on.
- The paper is somewhat well evaluated, with appropriate ablation studies.


Weaknesses:
- The method is evaluated on very synthetic data, well prepared for the model to disambiguate easily between object and background. There is a single object in each scene. This is a shared limitation of related projects in the literature, however, it is worth acknowledging, as this type of data is not common in the real world and therefore limit the scope of the "physically-embodided systems" the paper targets.
- The method is exclusively studied on opaque objects, with some to little metallicness on them and with a very simple reflectance model. While this is somewhat fine in the context of the literature, the fact that phenomena like transparency, anisotropy, subsurface scattering or translucency are not studied, strongly limit the scope upon which this model is learning to do general-purpose perception of objects.
- It would have been interesting to study the inheret uncertainty of capturing material properties from single images or from few-shot imagery, as in UMat: Uncertainty-Aware Single Image High Resolution Material Capture (CVPR) or Uncertainty for SVBRDF Acquisition using Frequency Analysis (SIGGRAPH). This line of research is very relevant for this paper and should be analyzed in the related work section. I believe that the paper overstates its contribution in the uncertainty-awareness side, as it does not evaluate this part of the problem, and ignores related work on it.
- The metrics used for this paper should be expanded. More perceptual metrics must be used to evaluate this method, including: LPIPS, FLIP, and ideally more render-aware metrics, as in Perceptual quality of BRDF approximations: dataset and metrics (EG).
- The paper relies on specific motions for it to work. The impact of this is not clear, and it may be the case that t will not generalize properly to new motions, or to moving light sources. This is an important limitation.
- I think the paper sometimes overuses forward references (eg in Line 35, there is a reference to Figure 7a). This reduces the quality of the reading experience.
- The choice of model design would benefit from additional explanation.
- It seems like the model struggles in the single-image example in very specific ways: The model either places most of the variation on the geometry, or on the albedo, leaving the other somewhat constant (Fig 7a). This is unexpected behaviour and creates ill-behaved probability distributions, which maybe be less than ideal for uncertainty quantification. Can the authors elaborate on this?

---

> ### Author Rebuttal · Authors · 2025-07-31
>
> We thank the reviewer for recognizing the importance of our proposed problem and the effectiveness of our framework demonstrated through evaluations. We address the concerns and questions below.
>
> ## Weaknesses
>
> **W1**
> > **The method is evaluated on very synthetic data; there is a single object in each scene. This is a shared limitation of related projects, however, it is worth acknowledging, as this type of data is not common in the real world.**
>
> We agree and will explicitly discuss this assumption about isolated objects in the final version. In practice, we find it convenient to simply use a segmentation model like SegmentAnything2 to extract foreground objects, as we did for the captured images and videos in Figures 7(a) and 7(b). Please also see our response to **Reviewer 3 - WscD - Q2** on applying the model to more complex scenes.
>
> **W2**
> > **The method is exclusively studied on opaque objects, with some to little metallicness on them and with a very simple reflectance model; Transparency, anisotropy, subsurface scattering or translucency are not studied.**
>
> Our choice of BRDF model includes spatially-varying specular, roughness and metallicness parameters in addition to texture, which is comparable to or more general than the models used in many prior works. For example, IID [35] and RgbX [57] only consider the roughness and metallicness parameters.
>
> That said, we agree that scaling up the model to consider more complex materials like transparency or anisotropy is an exciting direction for future work. We observe that datasets for real-world evaluation on objects that have more complex materials are currently scarce, so there is need for work in that direction, too.
>
> **W3**
> >  **It would have been interesting to study the inherent uncertainty of capturing material properties from single images or from few-shot imagery, as in UMat: Uncertainty-Aware Single Image High Resolution Material Capture (CVPR) or Uncertainty for SVBRDF Acquisition using Frequency Analysis (SIGGRAPH).**
>
> These two works are very different from ours because they assume the shape is known. In this scenario there is much less ambiguity and the output distributions tend to be unimodal. Our case is different because the output distributions can contain very distinct (e.g., concave/convex) modes. Also, our model can recover complex, non-planar geometries with self-occlusions, and it can handle global illumination effects like interreflections and ambient occlusion. We will add this discussion to the final version.
>
> **W4**
> > **The metrics used for this paper should be expanded in include perceptual metrics.**
>
> Thank you for the suggestion. For geometry, we believe that angular errors are better metrics than perceptual ones. For albedo, we included three metrics: SSIM, PSNR, and RMSE. Below we have also added LPIPS using the same protocol, which we will include in the final version. It shows the same overall trend, with our model outperforming the others and with ColorfulShading being the second best.
>
> Note that we have included the new Marigold-Albedo model in the table below, in line with our response to **Reviewer 2 - yKQP - W3**.
>
> ---
> *Albedo Estimation (MIT-Intrinsic)*
>
> |    Model   | IID (Mean Sample) | IID (Top 3) |  RgbX  | ColorfulShading | Marigold - Albedo |  Ours  |
> |:------------:|:-----------------:|:-----------:|:------:|:---------------:|:--------:|:------:|
> | LPIPS (↓)    |      0.163       |    0.199   | 0.166 |      0.139     |  0.149  | 0.110 |
> ---
>
> **W5**
>
> > **The paper relies on specific motions for it to work. It may be the case that it will not generalize properly to new motions, or to moving light sources.**
>
> We find experimentally that our model generalizes fairly well to motions that are different from the ones that are used for training. For instance, in the left video of Figure 6, the teapot rotates around a horizontal axis that is orthogonal to training motion, and yet the model does fine. (See the supplemental html for a visualization.) Additionally, our new results in the response to **Reviewer 3 - WscD - W1** assess generalization to random-axis object rotations and random translations, and our new results in the response to **Reviewer 2 - yKQP - W4** assess generalization to wide-baseline camera motion.
>
> We do not explore moving light sources since that is essentially photometric stereo, which is better suited to laboratory settings and has methods that are quite mature (e.g.,[*]). Instead we focus on object motion because it is similar to situations where a natural observer picks up an object and moves it around to better understand its shape and material. This type of motion also relates directly to prior perceptual studies like [24] and mathematical studies like [12].
>
> [*] Ikehata. “Scalable, Detailed and Mask-Free Universal Photometric Stereo.” CVPR 2023.
>
> **W6**
>
> > **I think the paper sometimes overuses forward references.**
>
> We will modify the text to improve clarity.
>
> **W7**
> > **The choice of model design would benefit from additional explanation.**
>
> Please see Section 3.2.2 for our discussion on the model design. Our design is primarily driven by two needs: (1) improving data and compute efficiency, which is often quite challenging for pixel-space diffusion models, and (2) creating a  unified architecture for shape and materials.
>
> Need (1) motivates our use of local shift-invariant attention modules and decoupled spatial and temporal attention at high resolutions. Need (2) motivates our introduction of channel-mixer modules to facilitate communication between shape and materials. Additionally, our use of temporal attention is motivated by previous mathematical studies (e.g., [12]) showing that local space-time patterns provide constraints between object motion and surface normals and curvatures.
>
> **W8**
> >  **It seems like the model struggles in the single-image example in very specific ways: it either places most of the variation on the geometry, or on the albedo, leaving the other somewhat constant (Fig 7a). This is unexpected and creates ill-behaved probability distributions, which maybe be less than ideal for uncertainty quantification.**
>
> There may be a misunderstanding here. Our model’s probability distributions over shape and materials in the single-image case are not ill-behaved. They simply represent the ambiguities that inherently exist. The distribution over shapes and materials from a single image is always multi-modal, because there is always at least the planar ‘postcard’ explanation in addition to the veridical one. And in many cases, like Figure 1(iii), there are additional modes in the output distribution that correspond to convex or concave interpretations which are both mathematically correct. (See [23] for a deeper dive on these cases.)
>
> Our goal is not to develop a quantitative measure of ‘uncertainty’ for these output multimodal distributions. Instead we aim to create an effective way to draw samples from them.
>
> Our experiments show that our model provides diverse samples when the input is ambiguous and has a multimodal output distribution, such as in the single-image cases. They also show that the very same model provides more accurate and less diverse samples when there is motion to help reduce the inherent ambiguity.
>
> ## Questions
>
> **Q1**
> >  **What is the impact of the input resolution? Can this model be used on higher resolution inputs?**
>
> Currently the model works for a fixed resolution of 256x256, similar to previous diffusion-based models. There is ongoing research toward applying diffusion models at resolutions that are higher than their training resolution, for example by applying a model to overlapping regions while applying spatial regularization [%]. This may be worth exploring in future work.
>
> Separately, our use of neighborhood attention with relative position embeddings (RoPE) [51] at the finer scales may allow other ways to apply our model to higher resolutions with the right architectural modifications. This may also be worth exploring in future work.
>
> [%] Bar-Tal et al. "MultiDiffusion: Fusing Diffusion Paths for Controlled Image Generation". ICML 2023.
>
> **Q2**
> >  **What motivated the choice of the Disney BRDF? Would other BRDFs work worse?**
>
> We chose the Disney BRDF because it accurately covers a wide range of materials, including plastics and metals, smooth and rough surfaces, and retroreflective materials. We think other BRDF models could also work, provided they capture a similarly broad range of reflectance phenomena.
>
> **Q3**
> > **Could the authors provide more insights on the choice of loss weighting coefficients?**
>
> Our weighting choices are (0.4, 0.4, 0.3) for (shape, albedo, reflectance), and they are motivated by literature in multi-task learning. Our work is closer to multi-task learning because unlike inverse-rendering, we have no pixel-space rendering loss from a forward rendering equation. So we think of the shape, albedo and reflectance as being different ‘tasks’.
>
> In general, one wants to assign slightly larger weights to  more difficult tasks and slightly smaller weights to  easier tasks.  We tried multiple weight strategies such as (1/3, 1/3, 1/3 ) or (0.7, 0.2, 0.1), and overall we found that assigning slightly higher weights to shape and albedo leads to faster convergence.
>
> **Q4**
> > **Could the model be more "active" in its exploration? This would better emulate how humans explore material appearance.**
>
> This is a great suggestion for future work. Indeed, we view our proposed framework as a crucial first step toward gaining the ability for more active, human-like perception. In contrast to deterministic  models, generative approaches like ours offer a natural way to assess ambiguity by sampling multiple hypotheses. This should eventually enable perception systems that can recognize when a perceptual distribution is not unimodal, and that can query more views or initiate other modalities like touch to reduce the ambiguity.

---

> > ### Comment · Reviewer_yYrM · 2025-08-04
> > **Response to rebuttal**
> >
> > I appreciate the rebuttal provided by the authors, which has address my concerns successfully.
> >
> > I have also read the other reviews and rebuttals, and I am increasing my rating to a 5. Some of the weaknesses I pointed out in my original review are not easily addressable in a revision but are still meaningful, which prevent me from increasing the score to a 6.
> >
> > However, I think this paper can be ready for publication if the changes promised to it are adequately incorporated into the final version.

---

### Official Review · Reviewer_WscD · 2025-07-02

**Clarity:** 4
**Significance:** 3
**Originality:** 3
**Rating:** 5
**Confidence:** 3

**Summary:**

This paper aims to predict the graphical parameters (for BSDF) and geometries (normal) from input RGB signals in a generative way. The authors propose to disentangle the ambiguity between geometry and appearance by including multi-views from differential motions, and achieve great performances in selected experiments.

**Questions:**

Apart from the weaknesses, I also have the following questions. Note that some questions are not aiming to challenge the quality of the paper, but just for better understanding the potential and working scheme of the method, so I don’t list them as weaknesses.
1. motion related:
    1. what is exactly the definition of ‘differential object motion’ in this paper? Is only the motion caused by camera movement considered?
    2. will the model work for deformable objects (could be slightly deformable)?
    3. will the model work for static object but moving light?
2. Can the authors show some results of directly working on complex scenes with multiple objects? This results may help us understand whether the mode tend to rely on global features (rely on the overall shape of objects) or local patterns. If latter, the model should work kinda reasonably for complex scenes.
3. Can the authors show the diversity change of 10 stochastic predictions from one case, of static vs moving?
4. I recommend the authors include some relighting results based on the decomposed materials and shapes, which can show the flexibility of the proposed methods for the downstream task and can also make the paper more attractive.

**Ethical Concerns:**

["NO or VERY MINOR ethics concerns only"]

**Final Justification:**

Thanks for the authors' for their detailed responses to my questions and concerns, and all of them are addressed. I have thoroughly read the rebuttal to my reviews and also to other reviews. I would say, although the authors are trying to show their model's generalizable ability to various cases, the usage of the model is still constrained to slight differential motion and object centric setting, but not for general motion complexity and normal daily scenes as in-the-wild videos. This is an apparent weakness proposed by all the reviewers.

However, I believe a good paper does not equal to a perfect model or a good product. I'd like to advocate the paper for the following reasons:

1. The basic concept of this paper is intuitive and instructive: the appearance and the geometry is ambiguous. Ambiguity means no single solution, so instead of fitting the model on a large datasest to get a single MLE guess, this work proposes to adapt their prediction to more observations from novel views, intuitively like human behavior. This is 'novel'.
2. The underlying working scheme - matching local feature patterns across views to make better predictions, instead of relying on global understanding - shows potential to gneralize the idea to much more complex future research. So I would suggest the authors put some effort in the next version to discuss more about this part. To be more specific, the motion among objects (either deformable or not), cameras, and lights (static or moving) should perform very similar in local-feature wise. Therefore, the pipeline could benefit future larger models a lot.

Based on the efforts and atitudes of the authors put on this rebuttal, I hold a positive opinion that they have a strong willing to include the experiments in the rebuttal in their final version, and are willing to include more discussion as listed above. So I will increase my Rating to 5 in my final recommendation. However, based on the unignorable weakness, I can't make a rating 6 recommendation.

**Limitations:**

The authors sufficiently discuss the limitations in the paper, and I agree with them.

**Quality:**

3

**Strengths And Weaknesses:**

I really like the idea proposed by this paper, that the shape and the material disentanglement should be tackled in a multi-view way. I conclude the strengths of this paper as follows:
1. this paper proposes to use multiple views from differential object motions to eliminate the ambiguity between object shapes and materials.
2. They use a unified framework to predict the shape and the material. Instead working as two separate branchs respectively, these two tasks benefits each other.
3. Well-presented paper with every statement supported by proper citations and very clear explanations, making readers easy to follow and understand.

As for the weaknesses, I will list the following:
1. As shown in the limitation section, the motion considered in this paper is simple. Although explicitly stated, this weakness is still unignorable. Since we cannot guarantee similar observations would still hold in more complex motion settings.
2. The framework asserts that without reconstructing the environment light will lead to a more modular and flexible decomposition (line78-79), but this assertion (actually a strong foundation for the implementation of this work) is not well supported by experiments.


Overall this is a good paper in my opinion. But since I am not primarily working on this field, so these weaknesses may not be that so important for this field, and also I will refer to other reviewers’ opinion to adjust my understanding.

---

> ### Author Rebuttal · Authors · 2025-07-31
>
> We thank the reviewer for affirming that the proposed ideas are interesting and the unified framework is effective. We also thank the reviewer for the thoughtful comments and questions, which we address below.
>
> ## Weaknesses
>
> **W1**
> > **The motion considered in this paper is simple and we cannot guarantee similar observations would still hold in more complex motion settings.**
>
> Our model actually performs reasonably well for motions that are different from the vertical-axis rotations used for training. For instance, in the left video of Figure 6, the teapot rotates around a horizontal axis that is orthogonal to training motion, and yet the model does fine. (See the supplemental html for a visualization.)
>
> To quantify this, we performed an additional simulation experiment using 174 unseen objects with randomized textures, each rendered under three different settings: (1) a vertical-axis rotation, as used for training; (2) a rotation about a randomly-chosen axis; and (3) a translation of up to 8% object size in a random direction. Note that (2) and (3) are very different from training, and that (3) is equivalent to having a static image with translational misalignments between frames.
>
> The numbers are provided below, and they show there is limited degradation from in-distribution motions to out-of-distribution ones. We attribute this to the fact that local shape and material constraints derive from the relative angles between rotation axis, normals and curvatures (see [12]), and that since we use a wide variety of shapes during training, the model sees many of these local relationships, each affected by a local translation that depends on the point’s 3D position with respect to the rotation axis.
>
> ---
>
> - **Out-of-distribution Differential Motion Experiment**
>
> |                   Scenario                   | Shape – Mean AE ↓ | Shape – Median AE ↓ | Albedo – SSIM ↑ | Albedo – RMSE ↓ |
> |:--------------------------------------------:|:---------------:|:-----------------:|:-------------:|:-------------:|
> | In-distribution vertical-axis rotation       |     12.267      |      10.415       |     0.868     |     0.051     |
> | Out-of-distribution random rotation          |     15.062      |      13.158       |     0.863     |     0.052     |
> | Out-of-distribution random translation       |     16.652      |      14.681       |     0.858     |     0.054     |
>
> ---
>
> **W2**
> > **The framework asserts that without reconstructing the environment light will lead to a more modular and flexible decomposition (line78-79), but this assertion is not well supported by experiments.**
>
> Thank you for the great question. Our choice of omitting explicit illumination recovery is  motivated by psychophysical studies that show that humans do not perform explicit lighting estimation, or at least delay it until after shape inference [#, %].
>
> We agree that this argument is qualitative and not based on controlled experiments, so we will adjust our language to reflect this.
>
> [#] Ostrovsky et al. “Perceiving illumination inconsistencies in scenes”, Perception, 2005.
>
> [%] Kunsberg and Zucker. “How shading constrains surface patches without knowledge of light sources.” SIAM Journal on Imaging Sciences, 2014.
>
> ## Questions
>
> **Q1-1**
> > **What is exactly the definition of ‘differential object motion’ in this paper? Is only the motion caused by camera movement considered?**
>
> Differential simply means the object motion is small. We use it to distinguish from wide-baseline images that are typically used for multiview reconstruction, and to connect with previous mathematical studies (like [12]) that use (continuous) differential equations. In practice in our data, we see typical displacements of tens of pixels as an object moves..
>
> While we only train using object motion, we find that our model generalizes fairly well to cases where the camera moves instead, even when the camera movements are over large, irregular baselines. See the results in our response to **Reviewer 2 - yKQP - W4** and results on the Stanford ORB dataset.
>
> **Q1-2**
> > **Will the model work for deformable objects (could be slightly deformable)?**
>
> We find that our model performs reasonably well for modest deformations, but when the deformations become very large, the model can no longer exploit correlations between frames and instead behaves similar or worse than when given static (i.e., zero motion) inputs. This is an interesting direction for future work and we will add it to the Conclusion and Limitations.
>
> **Q1-3**
> > **Will the model work for static object but moving light?**
>
> No it does not. This would be a typical photometric stereo setup, for which a number of different methods exist (e.g., Ikehata, CVPR 2023 [*]). Photometric stereo is useful in laboratory settings, but it is otherwise somewhat artificial. Instead, we consider a scenario that is closer to when a natural observer picks up an object and moves it around to better understand its shape and material. In this scenario, the lighting still moves relative to the object, but the object moves relative to the camera as well.
>
> [*] Ikehata, “Scalable, Detailed and Mask-Free Universal Photometric Stereo.” CVPR 2023.
>
> **Q2**
> > **Can the authors show some results of directly working on complex scenes with multiple objects? This results may help us understand whether the mode tend to rely on global features or local patterns. If latter, the model should work kinda reasonably for complex scenes.**
>
> Great question. The short answer is that the model makes effective use of local patterns and so handles complex scenes quite well. We attribute this to:
> - (1) our training assets including scenes with multiple objects and self-occluding object-parts (e.g., fruits on a plate) and
> - (2) our use of local, shift-invariant attention at high resolutions, which helps avoid overfitting to global object features.
>
> There is supporting evidence in Figure 4, where the images are very abstract and different from the training set, without occluding contours or other global cues. There is also evidence in Figure 5, where the plate of cookies contains multiple objects, and the braided sphere contains self-occluding object parts that move in different directions. (See also the extended video of the braided sphere in the supplemental html.)
>
> We will improve our discussion of this important point in the final version.
>
> **Q3**
> > **Can the authors show the diversity change of 10 stochastic predictions from one case, of static vs moving?**
>
> We do this in the left of Figure 6, where we visualize the embeddings of 100 predictions along with three examples for each of the static and moving cases. We will add more examples of the predictions to the final supplement, but overall they look fairly similar to the ones in Figure 6.
>
> **Q4**
> > **I recommend the authors include some relighting results based on the decomposed materials and shapes, which can show the flexibility of the proposed methods for the downstream task and can also make the paper more attractive.**
>
> We included some relighting examples in the right-most column of Figure 5, and we will add more examples to the final supplement, including for captured objects. Thanks for the suggestion.

---

> > ### Comment · Reviewer_WscD · 2025-08-02
> >
> > Thanks for the authors' for their detailed responses to my questions and concerns, and all of them are addressed. I have thoroughly read the rebuttal to my reviews and also to other reviews. I would say, although the authors are trying to show their model's generalizable ability to various cases, the usage of the model is still constrained to slight differential motion and object centric setting, but not for general motion complexity and normal daily scenes as in-the-wild videos.  This is an apparent weakness proposed by all the reviewers.
> >
> > However, I believe a good paper does not equal to a perfect model or a good product. I'd like to advocate the paper for the following reasons:
> > 1. The basic concept of this paper is intuitive and instructive: the appearance and the geometry is ambiguous. Ambiguity means no single solution, so instead of fitting the model on a large datasest to get a single MLE guess, this work proposes to adapt their prediction to more observations from novel views, intuitively like human behavior. This is **'novel'**.
> > 2. The underlying working scheme - matching local feature patterns across views to make better predictions, instead of relying on global understanding - shows **potential** to gneralize the idea to much more complex future research. So I would suggest the authors put some effort in the next version to discuss more about this part. To be more specific, the motion among objects (either deformable or not), cameras, and lights (static or moving) should perform very similar in local-feature wise. Therefore, the pipeline could benefit future larger models a lot.
> >
> > Based on the efforts and atitudes of the authors put on this rebuttal, I hold a positive opinion that they have a strong willing to include the experiments in the rebuttal in their final version, and are willing to include more discussion as listed above. So I will increase my **Rating** to **5** in my final recommendation. However, based on the unignorable weakness, I can't make a rating 6 recommendation.

---

### Official Review · Reviewer_yKQP · 2025-07-02

**Clarity:** 4
**Significance:** 2
**Originality:** 2
**Rating:** 4
**Confidence:** 3

**Summary:**

This paper presents a conditional video diffusion model that performs joint estimation of object shape and material from short RGB video clips capturing differential motion. Unlike prior works that focus on single images or isolated attributes, this paper tackles the ambiguity inherent in inferring both shape and material by training a conditional generative model that outputs pixelwise surface normals and material maps, conditioned on RGB frames. To overcome the high computational requirements of modeling high-resolution outputs, the method uses a hybrid architecture: full spatio-temporal attention is applied only at lower resolutions, while higher resolutions use convolutional blocks with separate local neighborhood and spatial attention modules. The paper argues that motion helps disambiguate appearance cues. The model is trained on a synthetic dataset of 100k video clips covering approximately 1100 3D models under diverse textures and lighting. Experiments demonstrate plausible shape and albedo recovery, multi-modal predictions on ambiguous inputs, and modest improvements over single-frame baselines.

**Questions:**

- Albedo estimation: Marigold reportedly can estimate albedo too. Why is this not reported or compared directly?
- Baseline video input: Is it possible to adapt single-frame models to accept multiple frames (e.g. by simple frame stacking) to provide a direct baseline for video conditioning? Qualitative figures for this would also help.
- Real videos: Could the authors show at least a few more real short videos with object motion to demonstrate robustness?

**Ethical Concerns:**

["NO or VERY MINOR ethics concerns only"]

**Final Justification:**

I have read the rebuttal and other reviewers’ comments. The authors have addressed my main concerns, in particular by providing a direct comparison to the newly released Marigold-Albedo model, where their approach shows clear improvements. They also added more real-world validation and clarified evaluation protocols, which alleviates my earlier reservations about fairness and completeness of comparisons. While the limitations regarding generalization to in-the-wild scenarios remain, I now consider the contributions (novel joint shape/material generative estimation from motion, solid technical design, and thorough ablations) to outweigh these concerns. I am therefore upgrading my recommendation to borderline accept.

**Limitations:**

yes

**Quality:**

4

**Strengths And Weaknesses:**

**Strengths:**
- Exceptionally good writing, figures, and presentation.
- Novel problem setting: Using short videos for conditioning shape and material estimation is novel and addresses ambiguities that single-view methods struggle with.
- Interesting architecture: The hybrid design (global transformer at low resolution, local convolution plus attention at high resolution, channel mixing for inter-attribute communication) is well-motivated and technically sound.
- Pixel-space diffusion: Performing diffusion in image space is a reasonable design choice to preserve fine details compared to latent diffusion pipelines (e.g. Marigold)
- Clear qualitative results: The paper includes meaningful visualizations showing how motion resolves ambiguities (e.g. shiny vs. painted textures) and how this results in multiple diverse samples.
- The paper and supplementary demonstrate careful ablations and explanations, including the impact of motion, channel mixing, and temporal consistency.

**Weaknesses:**
- Generalization. The method is trained from scratch on the dataset of 1100 3D models. This is in contrast to methods like Marigold, which are finetuned form rgb image diffusion, and are not limited in generalization to real-world datasets.
- Marginal gains vs. Marigold on shape: From the numbers in Table 1, the improvement over Marigold or StableNormal for normal estimation appears modest.
- No baseline for albedo: The paper claims better albedo estimation but does not clarify whether comparable models like Marigold or RgbX could also be extended to report the same metric under similar settings; most importantly, Marigold already supports albedo estimation.
- Limited real-world validation: Although the paper shows some real-world photos, these are too few. There is no demonstration of real captured videos to show whether the differential motion cues generalize beyond synthetic motion. This is important given the first concern regarding limited generalization due to training on synthetic dataset.
- Unfair comparison protocol: For stochastic methods, the paper averages the top 3 of 10 samples for its model but uses the mean prediction for the IID baseline. additionally reporting the same protocol (e.g. best-of-10 or top-3) would be fairer.

**Minor:**
Lines 37–41 claim that most recent methods are non-generative and over-smooth ambiguities. This feels overstated as several recent works do address stochastic outputs; the claim could be softened.

In summary, the main concern is the limited comparison to existing baselines, real-world generalization and potential unfairness in the experimental protocol. If the authors can clarify the comparison details, report or explain concerns regarding albedo baselines, and add more real-world results, this would make the contribution stronger and i would then recommend acceptance.

---

> ### Author Rebuttal · Authors · 2025-07-31
>
> We thank the reviewer for recognizing the novelty of our problem formulation and architecture and for providing detailed feedback. We have conducted additional studies to address the concerns and questions.
>
> ## Weaknesses
>
> **W1**
> > **Generalization. The method is trained from scratch on the dataset of 1100 3D models. This is in contrast to methods like Marigold, which are finetuned from image diffusion, and are not limited in generalization to real-world datasets.**
>
> Thanks for this question; we will add a discussion.
>
> Prior work [23] convincingly shows that Marigold fails to capture multimodal output distributions (e.g. concave/convex) for ambiguous test images and captured photos, such as those in Figure 1(iii) and Figure 4. That work also shows similar failures for other models that are fine-tuned from Stable Diffusion: The SD-based models provide unimodal distributions instead of proper multimodal distributions that reflect the inherent mathematical ambiguities and match multi-stability in humans.
>
> In contrast, our model captures the multimodality very well, thanks to its joint estimation of shape and materials, its pixel-space diffusion architecture that avoids a VAE, its novel spatio-channel attention mechanism, and its training from scratch. Together, these innovations provide the ability to avoid overcommitting to any single shape/material explanation when inputs are truly ambiguous.
>
> It is possible that a pre-trained foundation model could provide a shortcut to achieving results like our model, but this has yet to be demonstrated.
>
> **W2**
> > **Marginal gains vs. Marigold on shape: From the numbers in Table 1, the improvement over Marigold or StableNormal for normal estimation appears modest.**
>
> Yes, but note that the Marigold-e2e-ft and StableNormal models are trained to specialize in shape estimation from a single image, while ours provides both shape and materials, not just from a single image, but also from a motion video, where it can leverage powerful motion cues. Also, as described above, our model does better at modeling multi-modal ambiguity when the input is inherently ambiguous.
>
> **W3**
> > **The paper claims better albedo estimation but does not clarify whether comparable models like Marigold or RgbX could also be extended to report the same metric under similar settings; most importantly, Marigold already supports albedo estimation.**
>
> Section 4.1 shows comparisons of albedo accuracy using three strong baseline models (IID, ColorfulShading and RgbX) and three different metrics. We did not compare to Marigold-Albedo because it wasn’t public until May 15th 2025, the same week as this NeurIPS deadline. We have since tested the new Marigold-Albedo model using the same metrics (top 3 out of 10), and we found that its performance is similar to ColorfulShading and RgbX, i.e., below our model:
>
> |      Model       | SSIM ↑ | PSNR ↑ | RMSE ↓ | LPIPS ↓ |
> |:----------------|:------:|:------:|:------:|:-------:|
> | Marigold–Albedo  |  0.72  | 19.86  |  0.11  |  0.15   |
> | Ours             |  0.80  | 23.14  |  0.08  |  0.11   |
>
> Note that a major difference between these two models (Marigold-albedo and RgbX) and ours is that they do not perform joint estimation of shape and albedo.Marigold uses two separate checkpoints for shape and albedo, and  RgbX uses a text prompt to switch between the two outputs. This is very different from our approach that directly outputs both and allows them to interact during the denoising process. We believe this is important to our accuracy gains, and our results demonstrate that it is critical for capturing ambiguities.
>
> **W4**
> > **Limited real-world validation.**
>
> Please see our results on real videos in Figure 7(b). In addition, Figures 3, 7(a) all show results on real-world objects.
>
> We also further tested on Stanford ORB benchmark [*], a dataset of images of real-world objects, by feeding our model sets of three masked images from nearby camera views. The effective “motion” in this experiment is very different from our training set because
> - (i) the camera moves instead of the object, and
> - (ii) the camera positions are separated by wide, irregular baselines.
>
> Despite this, we find that our model achieves high accuracy, especially in comparison to the two strongest baselines, Marigold-e2e-ft for shape and ColorfulShading for albedo using the same input images.
>
> Below are the numbers for six of the ORB benchmark’s scenes. Like Table 1, they show competitive performance on shape and superior performance on albedo. *This is additional evidence that our model can generalize beyond the synthetic motion used for training.*
>
> ---
> **Stanford ORB Real-world Objects Results**
> - Shape Inference
>
> |       Shape (Median AE ↓)        | Blocks | Cactus |   Car   |   Cup   | Grogu | Teapot |
> |:-------------------:|:------:|:------:|:-------:|:-------:|:-----:|:------:|
> | Marigold-e2e-ft [46]     | 10.19  | 14.59  |  13.69  |  12.14  | 13.27 | 13.25  |
> | Ours                | 11.22  | 19.50  |  13.86  |   7.85  | 10.84 | 15.02  |
>
> - Albedo Inference
>
> |       Albedo (SSIM ↑ / PSNR ↑)        |    Blocks    |    Cactus    |      Car     |      Cup     |     Grogu    |    Teapot    |
> |:-------------------:|:------------:|:------------:|:------------:|:------------:|:------------:|:------------:|
> | ColorfulShading [10]    | 0.76 / 24.69 | 0.66 / 25.58 | 0.64 / 23.01 | 0.84 / 24.83 | 0.61 / 19.87 | 0.69 / 23.33 |
> | Ours                | 0.74 / 25.28 | 0.78 / 26.23 | 0.70 / 23.68 | 0.88 / 24.97 | 0.86 / 24.05 | 0.79 / 24.12 |
>
> ---
>
> To further demonstrate the generalization of our model to other kinds of motions, we also conduct quantitative experiments on unseen assets generated with out-of-distribution motions such as random rotations and translations. Please see results in **Reviewer 3 - WscD - W1**.
>
> [*] Kuang et al. “Stanford-ORB: A Real-World 3D Object Inverse Rendering Benchmark”. NeurIPS 2023.
>
> **W5**
> > **Unfair comparison protocol: For stochastic methods, the paper averages the top 3 of 10 samples for its model but uses the mean prediction for the IID baseline. Additionally reporting the same protocol would be fairer.**
>
> We use the mean prediction for the IID baseline following the authors’ recommended setting because it achieves the best performance in their original paper [35]. The evaluation results with the alternative top-3 out of 10 evaluation protocol are below. They are consistently worse than the mean prediction, consistent with the original paper’s claims. Nonetheless, we agree that including the same metric (top 3 out of 10) will make our description more complete. Thank you for the suggestion.
>
> |  IID [36] | SSIM ↑ | RMSE ↓ | PSNR ↑ | LPIPS ↓ |
> |:--------------------------------------------|:------:|:------:|:------:|:-------:|
> | Mean prediction out of 10 samples           |  0.61  |  0.13  | 18.19  |  0.16   |
> | Best 3 out of 10 samples |  0.44  |  0.15  | 17.01  |  0.20   |
>
> **W6**
> > **Lines 37-41 claim could be  softened.**
>
> Thank you for the suggestion. We will adjust the wording.
>
> ## Questions
>
> **Q1**
> > **Albedo estimation: Marigold reportedly can estimate albedo too. Why is this not reported or compared directly?**
>
> Please see the discussion and new results in our response to **W3**. We note that by NeurIPS policy any work published after March 1st, 2025 is considered contemporaneous, and that Marigold-Albedo was not released until May. Thank you for suggesting the comparison. We will include our new results in the final version.
>
> **Q2**
> > **Baseline video input: Is it possible to adapt single-frame models to accept multiple frames (e.g. by simple frame stacking) to provide a direct baseline for video conditioning?**
>
> Yes it is possible, and we included some examples (using StableNormal) in the last section of the supplemental html. As shown in those examples, the problem with simple stacking is that it leads to flickering due to the lack of temporal consistency. Without training additional temporal interactions such as 3D convolutions or temporal attention, stacked models struggle to effectively leverage the time-varying inputs.
>
> We will add more examples to the supplement, including with albedo.
>
> **Q3**
> > **Real videos: Could the authors show at least a few more real short videos with object motion to demonstrate robustness?**
>
> Sure, we can add more that complement the existing ones in Figure 7(b). Since we are not able to upload media with this rebuttal, we instead conducted a quantitative evaluation using real ‘videos’ created from the real wide-baseline images in the Stanford ORB dataset. See our response to **W4**. We will also add more real videos to the final supplement.

---

### Official Review · Reviewer_EEKb · 2025-07-03

**Clarity:** 3
**Significance:** 3
**Originality:** 3
**Rating:** 5
**Confidence:** 3

**Summary:**

This paper presents a novel conditional denoising-diffusion model for generating shape-and-material maps from short input videos of objects undergoing differential motions. The model, named U-ViT3D-Mixer, is designed to handle the ambiguity in perceiving shape and material from a single image by leveraging motion cues. It can produce diverse, multimodal predictions for ambiguous static observations and converge to more accurate explanations when objects move. The model is trained on synthetic object-motion videos and exhibits compelling emergent properties, such as generating plausible multimodal samples for ambiguous inputs and effectively utilizing differential object motion to resolve perceptual ambiguities. The paper also demonstrates the model's performance on real-world objects, showing high-quality shape-and-material estimates.

**Questions:**

1.How would the approach generalize to camera motion, translation, or longer or non-rigid motion sequences? Have you performed any pilot studies or experiments on more diverse video data?

2.Can the model be extended to handle more complex lighting conditions, such as those with multiple light sources or non-uniform lighting?

3.Would the authors consider adding error bars, confidence intervals, or repeated trials to the key quantitative results in Table 1 and 2 to strengthen claims of empirical improvement? This would clarify if improvements are significant and reproducible.


4.How does the model perform on objects with highly specular or transparent materials, which are known to be challenging for shape and material estimation?

5.What are the main failure or degradation modes when the model is used with cluttered backgrounds, occlusions, or novel, unmodeled materials/lighting?


6.What are the specific reasons for the model's improved performance on albedo estimation compared to existing models?

**Ethical Concerns:**

["NO or VERY MINOR ethics concerns only"]

**Final Justification:**

I appreciate the authors' rebuttal, which successfully addressed most of my concerns.

I also read the other comments and rebuttals and raised my rating to 5. However, I was unable to raise the rating to 6 because the authors' initial manuscript did not provide results on their model for multi-object scenes, which I believe are significant and important to the paper's approach.

**Limitations:**

Yes

**Paper Formatting Concerns:**

This paper does not have any formatting issues.

**Quality:**

3

**Strengths And Weaknesses:**

Strengths:

1.	The paper introduces a new method for joint shape and material perception using a conditional video diffusion model, which is a fresh perspective in the field of computer vision.

2.	Figures such as Figure 3 (benchmarking against baselines) and Figure 6 (disambiguation via motion, PCA of sample clusters) offer strong evidence of the approach’s claims, with informative visual interpretations.

3.	The model is benchmarked quantitatively and qualitatively against strong baselines for shape and albedo estimation on challenging datasets (Table 1, Figure 3), and extensive ablations (Table 2, Figure 5) illustrate the necessity of key architectural choices.


Weaknesses:

1.While results on synthetic and some real images are promising, the model and experiments focus exclusively on isolated objects with controlled backgrounds and restricted motion types. The generalizability to multi-object scenes, unknown backgrounds, or real-world video with complex movement is not demonstrated.

2.The evaluation protocol for ambiguous cases is largely qualitative, relying on visualizations or sample diversity. While these are compelling, there is little systematic, quantitative assessment of how well the sampled distributions align with human uncertainty, nor with established perceptual ambiguity benchmarks apart from visual comparisons. This leaves some doubt as to how reliably the model’s uncertainty calibration matches true perceptual ambiguities.

3.The quantitative benchmarks (Table 1, Table 2) lack error bars, confidence intervals, or significance testing. The justification is that ambiguity is intrinsic, but this omission limits the rigor and interpretability of reported improvements and comparisons to baselines. For instance, gains on DiliGENT shape benchmarks (Table 1) are small and would benefit from significance estimates.

4.Training the model requires significant computational resources, taking around 5 days on 4 H100 GPUs, which might be a barrier for some researchers.

5.The synthetic dataset, though large and diverse, may not fully cover the complexity of real-world bidirectional reflectance distribution functions (BRDFs) or complicated indoor/outdoor lighting. Although Section 4 makes an effort in randomizing environment maps and textures, real-world scenes may involve more challenging or unmodeled conditions not validated here.

6.The main baselines in Table 1 focus on single-image models or those specializing in either shape or material. There is limited direct comparison to models that address joint estimation or leverage motion—even if prior art is scarce, this could be more explicitly discussed as a limitation or open problem.

7.The model is designed for isolated objects, and extending it to handle multi-object scenes with natural backgrounds would require additional training and reasoning about grouping and compositions.

8.While efficient in parameters for the design, the paper does not provide a detailed analysis of inference/training speed, GPU/memory requirements, or direct comparisons of parameter count and runtime to established baselines, which is increasingly important in applied settings.

---

> ### Author Rebuttal · Authors · 2025-07-31
>
> We thank the reviewer for finding our method novel and the experiment results extensive and convincing.
>
> ## Weaknesses
> **W1**
> > **Generalizability to different motion types, multi-object scenes, unknown backgrounds, or real-world video is not demonstrated.**
>
> We currently focus on object-centered scenarios where the background is clean or has been segmented with an off-the-shelf model like SegmentAnything2. Scaling up to uncontrolled backgrounds and more complex and dynamic scenes would be great for future work. Our model handles multiple objects just fine, because the occlusions and disocclusions in those cases are similar to the ones between object-parts (e.g., the braided sphere in Figure 5 and supp.).
>
> We also empirically find that our model can generalize reasonably well to motions different from the ones used for training. See the additional synthetic and captured results in our responses to **Reviewer 3 - WscD - W1** and **Reviewer 2 - yKQP - W4**.
>
> **W2**
> > **Quantitative assessment with established perceptual ambiguity benchmarks.**
>
> Comparison with human perceptual ambiguity is indeed an interesting avenue of future work. There are no established human perceptual ambiguity benchmarks related to joint material and shape perception, partly because before our work there were no computational models capable of capturing shape and material ambiguities like the one demonstrated in [24]. We hope that papers like ours will help inspire future work in psychophysics and inspire joint human and computational perception studies.
>
> **W3**
> > **Quantitative benchmarks lack error bars, confidence intervals, etc.**
>
> To further support the quantitative evaluations, we repeat the same experiments for our models for 5 runs and report the mean and standard deviations here. Note that the best-performing shape baseline (*Marigold-e2e-ft*) and albedo baseline (*ColorfulShading*) are deterministic so no error bars are reported. We see consistent performance as Table 1 in the main paper. For Table 2 ablations, the results also confirm the effectiveness of proposed modules.
>
> ---
>
> *Table 1*
> |       Method       | Mean AE | Median AE |  SSIM | RMSE | PSNR |
> |:------------------:|:---------------:|:-----------------:|:-------------:|:-------------:|:-------------:|
> |        Ours        | 17.782 ± 0.292  | 11.890 ± 0.418    | 0.816 ± 0.008 | 0.076 ± 0.002 | 23.817 ± 0.119 |
>
> *Table 2 -with motion*
>
> | Model                                 |     Mean AE ↓     |      SSIM ↑      |
> |:-------------------------------------|:----------------:|:---------------:|
> | U-ViT3D (Base)                           | 18.899 ± 0.324   | 0.775 ± 0.003   |
> | + Local Spatial-Temporal Attention     | 14.688 ± 0.203   | 0.826 ± 0.003   |
> | + Multi-Attribute Mixer                 | 11.832 ± 0.241   | 0.841 ± 0.002   |
>
> ---
>
> **W4**
> > **Training requires significant compute (~ 5 days on 4 H100 GPUs).**
>
> The computational resource required by our model is modest compared with other trained-from-scratch or finetuned generative models. For comparison, the IID model [35] requires ~24 A6000 GPU days for finetuning and the Diffusion Forcing Transformer [49] requires 60 H100 GPU days to train a video generation model at the same 256 by 256 resolution.
>
> Our architecture and training protocol provide a general template for creating generative models from scratch for tasks that require leveraging novel datasets or modalities and that do not have a suitable pre-trained foundation model or VAE to start from.
>
> **W5**
> > **The synthetic dataset may not fully cover complex real-world BRDFs or lighting.**
>
> Our training data exhausts the parameter space of the analytical BRDF model we adopt, heavily samples the space of environment-map illuminations, and includes global lighting effects like inter-reflections and ambient occlusion. This actually does cover real-world materials and shape, as demonstrated by the accuracy of the model on real-world objects.  For instance, the model generalized to laboratory point light sources in the DiLiGent dataset (as shown in Figure 3) as well as captured real-world videos in Figure 7(b). This ability to fully leverage synthetic data for problems in which real-data supervision is near impossible to obtain is one of the contributions of our work.
>
> **W6**
> > **The main baselines in Table 1 focus on single-image models or those specializing in either shape or material. There is limited direct comparison to models that address both or leverage motion.**
>
> To the best of our knowledge, our work is the first to perform simultaneous estimation of  shape and material, for either static or moving objects. As such, there are no methods we can directly compare with. Instead, we compared the best shape or material models, and we included variations of our full model as comparisons in Table 2 and Figure 5 to show the effectiveness of the model’s components. We will clarify this in the paper.
>
> **W7**
> > **Extension to handle multi-object with natural backgrounds.**
>
> Our model handles multiple objects just fine (please see response to **W1**). We will show more results and discuss them in the final version. Natural backgrounds should not be a problem either as long as off-the-shelf segmentation methods work. We will test and add results.
>
> **W8**
> > **Comparison to baselines on GPU requirements and inference speed.**
>
> Training and inference speed are discussed in the paper’s Section 4 under *Implementation*. To supplement this, the following compares to other stochastic, diffusion based models. We use each model’s FP32 version conduct the analysis on a single A100 GPU.
>
> This shows that our model is compact, fast and memory efficient, due to its architectural design.
>
> |         Model         |               Output               | Model Size (FP32) | Inference Time | Inference GPU Memory |
> |:---------------------|:----------------------------------|:-----------------:|:--------------:|:--------------------:|
> | Ours                  | Shape and Materials, three frames |      0.4 GB       |     2.7 s      |       2.8 GB         |
> | StableNormal    | Shape only, single-frame           |      3.8 GB       |     1.1 s      |      19.5 GB         |
> | IID             | Materials only, single-frame       |      6.2 GB       |     6.0 s      |       9.5 GB         |
> | RgbX             | Shape or Materials, single-frame   |      3.8 GB       |    10.5 s      |       6.0 GB         |
>
> ## Questions
>
> **Q1**
> > **How would the approach generalize to camera motion, translation, or longer or non-rigid motion sequences?**
>
> - For out-of-distribution object motions like random rotation or translation, please see the quantitative results on synthetic data in **Reviewer 3 - WscD - W1**.
>
> - For real-world camera motions that consist of both rotation and translation, please see the results on real-world captures in **Reviewer 2 - yKQP - W3**.
>
> - For longer motion sequences, please see the supplementary material, where our model provides better temporal consistency and higher quality  than the baseline (StableNormal [56]).
>
> - For non-rigid motions, our model performs reasonably well for modest deformations, but when the deformations become very large, the model behaves similar or worse than when given static inputs. Handling deformation is an interesting direction for future work. We will add discussion.
>
> **Q2**
> > **Can the model be extended to handle more complex lighting conditions?**
>
> These are already in the test set (e.g., Figure 5 where specular objects are rendered under complex illumination). Our model accurately estimates the shape and material of objects captured under such complex, multi- and/or non-uniform lighting. The supplementary material also includes such instances.
>
> Figure 7 (b) shows results on objects taken under complex lighting conditions with multiple light sources both from indoor lighting and outdoor natural light. We will clarify by discussing these results in terms of lighting complexity.
>
> **Q3**
>
> > **Error bars, repeated trails for key results.**
>
> See **W3**.
>
> **Q4**
>
> > **Performance on objects with highly specular or transparent materials.**
>
> The model does great for any sort of opaque object we have tested. For examples, please see Figure 1(ii) and Figure 5.
>
> Our current model produces a flat ‘postcard’ explanation for transparent objects. Extending our model to handle transparency and translucency are our next targets for future work, and they are interesting in terms of human perception as well (see [#]).
>
> [#] Marlow and Anderson. "The Cospecification of the Shape and Material Properties of Light Permeable Materials", PNAS 2021.
>
> **Q5**
>
> > **Main failure mode**
>
> Our model does well for opaque objects under any environment lighting. For transparency it breaks down as described previously.
>
> Our model also handles occlusions well, either between the parts of a single object or between different objects. For example, the plate of cookies in Figure 5 contains multiple objects, and the braided sphere in that figure contains occlusions between object parts that are moving in opposite directions (please see the extended video results on the supplemental html page).
>
> **Q6**
>
> > **Reasons for improved performance on albedo estimation.**
>
> Modeling in pixel space instead of a latent space (VAE-based) is important, as well as using randomized textures during training.
>
> Joint training with shape also plays a significant role. As demonstrated in the paper, inferring albedo alone can be quite ambiguous, especially from a single view. It is possible that shape-domain supervision helps encourage the model to learn better priors on shape AND albedo, allowing it to leverage the learned shape prior to jointly sample plausible solutions.  This is in line with prior studies like [*] that show the positive confluence of joint training for multiple vision tasks, especially when data is scarce.
>
> [*] Misra et al. “Cross-stitch Networks for Multi-task Learning”. CVPR 2016.

---

> > ### Comment · Reviewer_EEKb · 2025-08-05
> >
> > I appreciate the authors' rebuttal, which successfully addressed most of my concerns.
> >
> > I also read the other comments and rebuttals and raised my rating to 5. However, I was unable to raise the rating to 6 because the authors' initial manuscript did not provide results on their model for multi-object scenes, which I believe are significant and important to the paper's approach.

---

### Author Response · Authors · 2025-08-07

We thank all reviewers for their attention and thoughtfulness. The feedback has helped strengthen the paper. We will incorporate the rebuttal experiments and discussions into our paper.

---

### Decision · Program_Chairs · 2025-09-17

**Decision:**

Accept (poster)

**Comment:**

The paper proposed a conditional diffusion model to estimate the shape and material from a short video of object motions. The reviewers like the proposed idea which is novel and well-presented. The proposed model has also undergone careful ablation. But they also suggested more evaluation on multi-objects and real-world settings. The rebuttal provided detailed clarifications and many additional results. It resolved most of the reviewers’ concerns and made most of the reviewers increase their ratings. Now, all the reviewers expressed a positive attitude toward the paper. The AC concur with the reviewers and believe that the proposed new model could be valuable for the vision and graphics community. As suggested by the reviewers, the final version should include the results in the rebuttal and provide more discussion of the model, especially its generalization ability on the real-world settings.